# Single cell RNA-seq and ATAC-seq analysis of cardiac progenitor cell transition states and lineage settlement

Guangshuai Jia[1], Jens Preussner[1,2], Xi Chen[3], Stefan Guenther[1,2], Xuejun Yuan[1,2], Michail Yekelchyk[1,2], Carsten Kuenne[1,2], Mario Looso[1,2], Yonggang Zhou[1], Sarah Teichmann [3,4,5] & Thomas Braun[1,2]

Formation and segregation of cell lineages forming the heart have been studied extensively but the underlying gene regulatory networks and epigenetic changes driving cell fate transitions during early cardiogenesis are still only partially understood. Here, we comprehensively characterize mouse cardiac progenitor cells (CPCs) marked by *Nkx2-5* and *Isl1* expression from E7.5 to E9.5 using single-cell RNA sequencing and transposase-accessible chromatin profiling (ATAC-seq). By leveraging on cell-to-cell transcriptome and chromatin accessibility heterogeneity, we identify different previously unknown cardiac subpopulations. Reconstruction of developmental trajectories reveal that multipotent Isl1$^+$ CPC pass through an attractor state before separating into different developmental branches, whereas extended expression of *Nkx2-5* commits CPC to an unidirectional cardiomyocyte fate. Furthermore, we show that CPC fate transitions are associated with distinct open chromatin states critically depending on *Isl1* and *Nkx2-5*. Our data provide a model of transcriptional and epigenetic regulations during cardiac progenitor cell fate decisions at single-cell resolution.

[1] Department of Cardiac Development and Remodeling, Max Planck Institute for Heart and Lung Research, 61231 Bad Nauheim, Germany. [2] German Centre for Cardiovascular Research (DZHK), Partner site Rhein-Main, Frankfurt am Main 60596, Germany. [3] Wellcome Trust Sanger Institute, Wellcome Trust Genome Campus, Hinxton, Cambridge CB10 1SA, UK. [4] EMBL-European Bioinformatics Institute, Wellcome Trust Genome Campus, Hinxton, Cambridge CB10 1SD, UK. [5] Theory of Condensed Matter, Cavendish Laboratory, 19 JJ Thomson Ave, Cambridge CB3 0HE, UK. These authors contributed equally: Guangshuai Jia, Jens Preussner, Xi Chen. Correspondence and requests for materials should be addressed to T.B. (email: thomas.braun@mpi-bn.mpg.de)

Cell fate mapping experiments demonstrated that cardiac progenitor cells (CPCs) in the mouse form from Mesp1+ cells that leave the primitive streak during gastrulation at E6.5 (reviewed in ref. [1]). At E7.5, CPCs express the homeobox genes Nkx2-5, Isl1, or a combination of both and exhibit a multilineage potential enabling them to generate cardiomyocytes, smooth muscle cells, endothelial cells, and pericytes[2,3]. During early developmental stages CPCs are located in two distinct anatomical locations, the first (FHF) and second heart field (SHF) [4–6]. Unlike FHF cells, SHF cells show delayed differentiation into myocardial cells and represent a reservoir of multipotent CPCs during cardiogenesis[1]. Isl1 is primarily expressed in CPCs of the SHF, making the Isl1[nGFP/+] knock-in reporter mouse line a reliable source for isolation of SHF cells[7,8]. In contrast, Nkx2-5 expression marks cells of both the FHF and SHF including the cardiac crescent and the pharyngeal mesoderm[1,9,10]. Although transient co-expression of Isl1 and Nkx2-5 has been observed, several lines of evidence indicate that Isl1 and Nkx2-5 suppress each other thereby allowing expansion of Isl1+ CPCs and differentiation into Nkx2-5+ cardiomyocytes[8,9].

Differentiated cells (e.g. cardiomyocytes) are assumed to acquire their identity in a successive step-wise manner from multipotent cells (e.g. CPCs) but the different intermediate states allowing transition from multipotent precursor cells to differentiated descendants still await further characterization. Global analysis of transcriptional changes does not provide the resolution for precise identification of such specific cellular transition states. Recent advances in single-cell RNA sequencing (scRNA-seq) permit characterization of transcriptomes at the single cell level at multiple time points, thereby allowing detailed assessment of developmental trajectories of precursor cells[11]. Single cell ATAC-seq (assay for transposase-accessible chromatin using sequencing) offers a similar power of resolution and generates additional information about gene regulatory processes[12,13]. However, bulk or single cell ATAC-seq have not yet been applied to characterize chromatin accessibility and putative regulatory elements driving cardiogenesis.

Here, we use scRNA-seq to transcriptionally profile FACS-purified Nkx2-5+ and Isl1+ cells from E7.5, E8.5 and E9.5 mouse embryos. We decided to focus on native embryonic cells and not on ESC derivatives, since some in vitro results have to be viewed with caution despite some advantages of ESC-based approaches[14,15]. By taking advantage of unsupervised bioinformatics analysis, we reconstruct the developmental trajectories of Nkx2-5+ and Isl1+ cells and identified a transition population in Isl1+ CPCs, which become developmentally arrested after inactivation of Isl1. Furthermore, we show that the transcriptional heterogeneity of Isl1+ CPCs is reflected by chromatin openings in individual Isl1+ CPCs at E8.5 and E9.5. We demonstrate by scRNA-seq and chromatin accessibility mapping that forced expression of Nkx2-5 is associated with de novo chromatin opening and primes the cardiomyocyte fate.

## Results

**Single cell transcriptomics of cardiac progenitor cells.** To unravel the molecular composition of either Isl1+ or Nkx2-5+ CPCs, we isolated GFP+ cells by FACS from Nkx2-5-emGFP transgenic and Isl1[nGFP/+] knock-in embryos (Fig. 1a) at E7.5, E8.5, and E9.5 and performed single-cell RNA sequencing using the Fluidigm C1 workstation (Fig. 1b). Insertion of the GFP-reporter gene into one allele of the Isl1 gene had measurable effects on Isl1 expression levels but caused no apparent defects during cardiac development and in adult stages[8]. The Nkx2-5-emGFP transgenic mouse line was generated using a BAC containing both the promoter region and distal regulatory elements,

which enables faithful recapitulation of Nkx2-5 expression[7]. After removal of low-quality cells (Supplementary Fig. 1a–g), we obtained 167 Nkx2-5+ and 254 Isl1+ cell transcriptomes, which cover most stages of early heart development (Fig. 1b).

We first asked whether Nkx2-5+ and Isl1+ CPCs sampled at successive developmental time-points are composed of distinct subpopulations. Therefore, we analyzed the coefficient of variation and dropout rates to defined heterogeneous genes as input for a neuronal network-based dimension reduction strategy (self-organizing map, SOM)[16], (Supplementary Fig. 2a, b). Projection of the resulting SOM into 2D for visualization by t-distributed stochastic neighbor embedding (t-SNE) identified three major subpopulations of Nkx2-5+ and five subpopulations of Isl1+ cells (Fig. 1c). The Nkx2-5+ cluster 3 mainly comprised E7.5 cells, whereas cluster 1 contained cells from E8.5 and E9.5 implying an intermediate cell state. Cluster 2 predominantly contained cells from E9.5 (Fig. 1d). Stage-dependent clustering was less evident for the five Isl1+ subpopulations, which might indicate that the specific cellular phenotypes of Isl1+ subpopulations are maintained for longer time periods (Fig. 1d).

To identify cluster-specific, differentially expressed genes, we used MAST[17] and a gene ranking approach implemented in SC3[18]. The top 269 and 216 genes that were differentially expressed in the Nkx2-5+ and the Isl1+ lineage, respectively, included several established cardiac regulatory genes such as Hand1, Tbx3/4/5, Gata2/3, Smarcd3, Rbm24, Wnt5a, Bmp4, Notch1, and Fgf3/15 (Fig. 1e; Supplementary Data 1, 2)[19–23]. Importantly, we detected numerous differentially expressed genes that so far had not been linked to cardiogenesis probably due to restricted expression in a small number of cells (Fig. 1e; Supplementary Data 1, 2). For example, Isl1+ cluster 5 expressed the cardiac transcription factors (TFs) Tbx3/4 and Wnt5a as well as several posterior Hox genes including Hoxa7/9/10, Hoxb6, Hoxc8, and Hoxd8 (Fig. 1e; Supplementary Fig. 3a). In addition, we newly identified several TFs such as Sox7/18, Sall3, Zbtb20, Zfp462/512b/711, Klf14; G-proteins including Arhgap1, Adgrf5, Arhgef15; Polycomb group (PcG) member Asxl3, the de novo DNA methyltransferase Dnmt3b in Nkx2-5+ or Isl1+ clusters (Fig. 1e; Supplementary Data 1, 2).

Next, we assigned identities to each cluster based on the expression of key marker genes (Fig. 1f, g). Consistent with the gene ontology analysis of differentially expressed genes within each cluster, Nkx2-5+ cluster 2 and Isl1+ cluster 2, which are enriched for GO terms such as muscle contraction and are characterized by cTnt and α-smooth muscle actin expression, appear to represent a myogenic fate, whereas Isl1+ cluster 1, which expresses Cd31 and is enriched for GO terms related to endothelial cell differentiation, is characterized by endothelial cell features (Fig. 1f, g; Supplementary Fig. 4a, b). Interestingly, expression of Nkx2-5 and Isl1 varies among subpopulations within each lineage: (i) Nkx2-5 shows more pronounced expression in late stages (clusters 2 and 3, E8.5 and E9.5) (Figs. 1f, 2e); (ii) Isl1 expression decreases in the cells expressing differentiation markers (clusters 2 and 1) (Figs. 1g, 2f). In addition to numerous differentially expressed genes, all three clusters of Nkx2-5+ CPCs show an enrichment of GO terms related to muscle development and contraction (Supplementary Fig. 4b) suggesting that Nkx2-5 is associated with myogenic differentiation while Isl1 is linked to the maintenance of progenitor cell multipotency.

To test the robustness of our approach and to analyze whether sufficient numbers of cells were sequenced to unveil the entire heterogeneity of CPCs, we generated single-cell transcriptomes of additional 663 Nkx2-5+ CPCs using the WaferGen iCell8 system (Fig. 1b). After correction of batch effects[24], merging and aligning

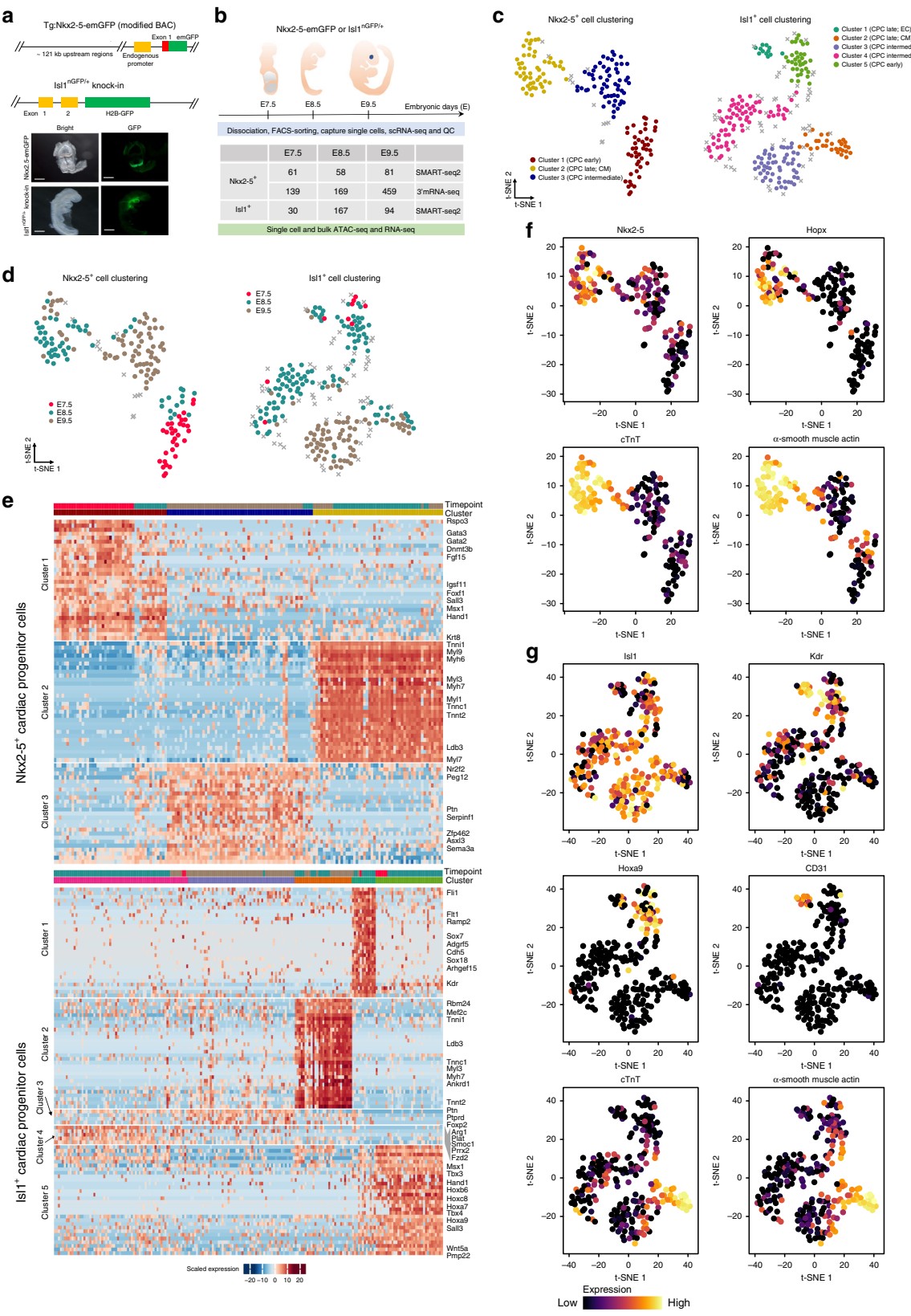

of data from both approaches (Supplementary Fig. 5a, b), we essentially mirrored the C1 data (Supplementary Fig. 5c, d). In particular, we detected a similar distribution of marker genes in cluster 1 and 3 but did not fully reproduce the marker gene pattern for cluster 2 (Supplementary Fig. 5e). We concluded that

sequencing depth rather than cell numbers is the main limiting factor for the discovery of novel genes in CPCs. Thus, we hereafter focused our analysis on the C1 data, which provided substantially deeper sequence coverage (Supplementary Fig. 1a–e).

**Fig. 1** Identification of CPC subpopulations by single-cell RNA-seq. **a** Schematic representation of the Nkx2-5-emGFP transgenic reporter and Isl1[nGFP/+] allele (top). Expression of Nkx2-5-emGFP and Isl1-nGFP at E8.5 in mouse embryonic hearts. (bottom). **b** Sampling time points for scRNA-seq, bulk RNA-seq, scATAC-seq, and bulk ATAC-seq. The table shows numbers of cells used for scRNA-seq. QC: quality control. **c, d** t-SNE visualization of individual Nkx2-5[+] and Isl1[+] CPCs to identify subpopulations. Colors denote corresponding clusters, and (**d**) development stages. Outlier cells are indicated by gray crosses. **e** Hierarchical clustering of expression heatmaps showing differentially expressed marker genes (AUROC > 0.8, FDR < 0.01; and lower bound of LogFC > 2 or higher bound of LogFC < −2, FDR < 0.01) across different clusters in Nkx2-5[+] CPCs (top) and Isl1[+] CPCs (bottom). Source data are provided in the Source Data file. **f, g** Expression of selected individual genes in Nkx2-5[+] (**f**) and Isl1[+] (**g**) CPCs. The colors represent expression levels of cells that are shown in the t-SNE plots in (**c**). EC, endothelial cell. CM, cardiomyocyte. Scale bar: 300 μm

**Reconstruction of development trajectories of CPCs**. scRNA-seq data allow ordering of cells by pseudotime based on cell-to-cell transcriptome similarity for calculation of developmental trajectories. We mapped cells collected at successive developmental stages along the pseudotime for reconstruction of the developmental trajectories of Nkx2-5[+] and Isl1[+] CPCs (Fig. 2a, b). Interestingly, cells collected at the same embryonic stages aligned to broad pseudotime points, suggesting that CPCs progress differentially through the developmental program. Cells undergoing a critical fate decision (such as lineage bifurcation) have been postulated to pass transition states[25] corresponding to a switch between different attractor states[26]. To delineate such transition states, we calculated the critical transition index of Nkx2-5[+] and Isl1[+] cell clusters [abbreviated as $I_c(c)$][27]. The $I_c(c)$ values of Nkx2-5[+] clusters showed similar numerical ranges essentially excluding the existence of transition states in the Nkx2-5[+] cell population. Instead, cells from later stages (cluster 1) showed decrease of $I_c(c)$ indicating stable settlement into an attractor state (Fig. 2c, left) with cardiomyocyte-like expression characteristics (Fig. 1e). In contrast, computation of $I_C(c)$ values of Isl1[+] clusters revealed decreased values for cells at the bifurcation point (cluster 3 and 4). Cells that overcame this point (clusters 1 and 2) exhibited more coordinated changes of gene expression changes (Fig. 2c, right). Since the critical transition index only hints to the presence of a transition state but does not reveal its stability, we calculated pairwise cell-to-cell distances (Supplementary Fig. 6a, b)[28]. As expected, cell-to-cell distances of Nkx2-5[+] CPCs did not change dramatically while cell-to-cell distances of Isl1[+] CPCs in cluster 3 and 4 increased substantially (Fig. 2d) indicating that Nkx2-5[+] CPCs follow one continuous trajectory without distinct transition states (Fig. 2a). In contrast, the trajectory of Isl1[+] CPCs bifurcated into two distinct orientations (endothelial cells and cardiomyocytes), suggesting the existence of a transition state with elevated noise levels, separating multipotency of Isl1[+] CPCs from acquisition of distinct cellular identities (Fig. 2b).

To identify genes potentially required to determine and/or maintain corresponding cell states, we generated a list of 108 genes for Nkx2-5[+] cells and 130 genes for Isl1[+] cells positively correlated with progression of pseudotime across clusters (Spearman rank correlation coefficient >0.5; Supplementary Fig. 7a, b; Supplementary Data 3) focusing on TFs and their chromatin-modifying partners (Fig. 2e, f). Genes expressed highly at early stages of the developmental trajectory were assigned to a priming category. Genes expressed at fate-restricted stages but not in multipotent progenitor cells were placed in a de novo category (Supplementary Fig. 7c, d). We noted that expression of *Dnmt3b*, *Gata2/3*, *Hand1*, and *Msx1* declined along Nkx2-5[+] developmental trajectories, qualifying them as priming genes. In contrast, increased expression of *Nkx2-5*, *Ankrd1*, *Cdkn2d*, *Hopx*, *Mef2c*, *Myocd*, *Smyd1*, *Tgfb1i1*, and *Tbx20* during Nkx2-5[+] CPC differentiation qualified them as de novo genes (Fig. 2e). Priming genes for Isl1[+] CPCs included TFs of the *Hox*-family, *Gata2*, *Hand1*, and *Tbx3/4* as well as *Hhex*, *Msx1*, *Sall4*, and *Snai1*. *Mef2c*, *Nkx-5*, *Tgfb1i1*, *Ankrd1*, *Myocd*, and *Smyd1* were

expressed at fate-restricted stages and hence represent de novo genes (Fig. 2f). The distinct expression pattern of priming and de novo TFs in Nkx2-5[+] compared to Isl1[+] cells indicates that the fate of progenitor cells is governed by different gene regulatory networks compared to fate-restricted cells.

**Comparison of Isl1[+] and Nkx2-5[+] cardiac progenitor cells**. The pseudotime analysis revealed that Isl1[+] and Nkx2-5[+] CPCs show different trajectorial patterns. On the other hand, it is known that some cells during cardiac development transiently co-express *Isl1* and *Nkx2-5*[9,29,30]. To investigate *Isl1* and *Nkx2-5* co-expressing cells more closely, we took advantage of the Isl1[+/nGFP] and Nkx2-5-emGFP reporter alleles, whose gene products are located in different subcellular compartments. Analysis of Isl1[+/nGFP]/Nkx2-5-emGFP compound embryos showing the expected intracellular GFP distribution in isolated CPCs (Fig. 3a) revealed that at E8.5 the majority of GFP[+] CPCs expressed *Isl1* (61 ± 3.8%) but not *Nkx2-5*. 21 ± 0.4% expressed *Nkx2-5* but not *Isl1*, while the remaining 15 ± 2.7% co-expressed both genes (Fig. 3b). The relative proportion of Isl1[+]Nkx2-5[−] cells (41% ± 1.5) declined at E9.5 accompanied by an increase of Isl1[−]/Nkx2-5[+] (28% ± 5.1) and Isl1[+]/Nkx2-5[+] cells (29% ± 3.8) (Fig. 3b). Consistently, Isl1[+] and Nkx2-5[+] cells showed a spatial overlap both at E8.5 and E9.5 (Supplementary Fig. 8).

Interestingly, analysis of scRNA-seq profiles indicated that *Isl1* and *Nkx2-5* co-expressing cells do not form a distinct group within the Nkx2-5[+] CPC population (Fig. 3c). In contrast, *Isl1* and *Nkx2-5* are co-expressed in cluster 2 (CPC late, CM) of the Isl1[+] CPC populations (Fig. 3c), which corresponds to the position of *Nkx2-5* expressing cells on the Isl1 pseudotime trajectory lineage (Fig. 3d). Isl1 expression was primarily found at early stages of the *Nkx2-5* trajectory, albeit expression was rather low, suggesting that strong expression of *Nkx2-5* at early stages might already cause a strong cardiomyocyte commitment (Fig. 3d, e).

Since the pseudotime analysis suggested different roles of CPCs expressing either high levels of *Isl1* or *Nkx2-5*, we directly compared transcriptional profiles of the Isl1[+] vs. the Nkx2-5[+] lineage at E8.5 and identified several differentially expressed genes (Fig. 4a). In agreement with the scRNA-seq data (Fig. 4a), *Ankrd1* was co-expressed with *Nkx2-5* in looping heart tubes by RNA in situ hybridization (Fig. 4b). In situ hybridization also confirmed moderate to high expression levels of *Sall3* and *Sox18* in the Isl1[+] lineage, which was not seen in the Nkx2-5[+] lineage (Fig. 4c). Other examples included *Asxl1* and *Msx1*, whose low expression levels by scRNA-seq analysis in the Isl1[+] but not the Nkx2-5[+] lineage was corroborated by RNA in situ hybridization (Fig. 4d).

**Isl1 is indispensable for CPC fate bifurcation**. The loss of *Isl1* results in absence of outflow tract and right ventricle and early embryonic lethality[31], which prevents dissection of *Isl1* dependent molecular processes in the SHF. To address the role of *Isl1* in cell fate determination, we inactivated the *Isl1* gene by generating

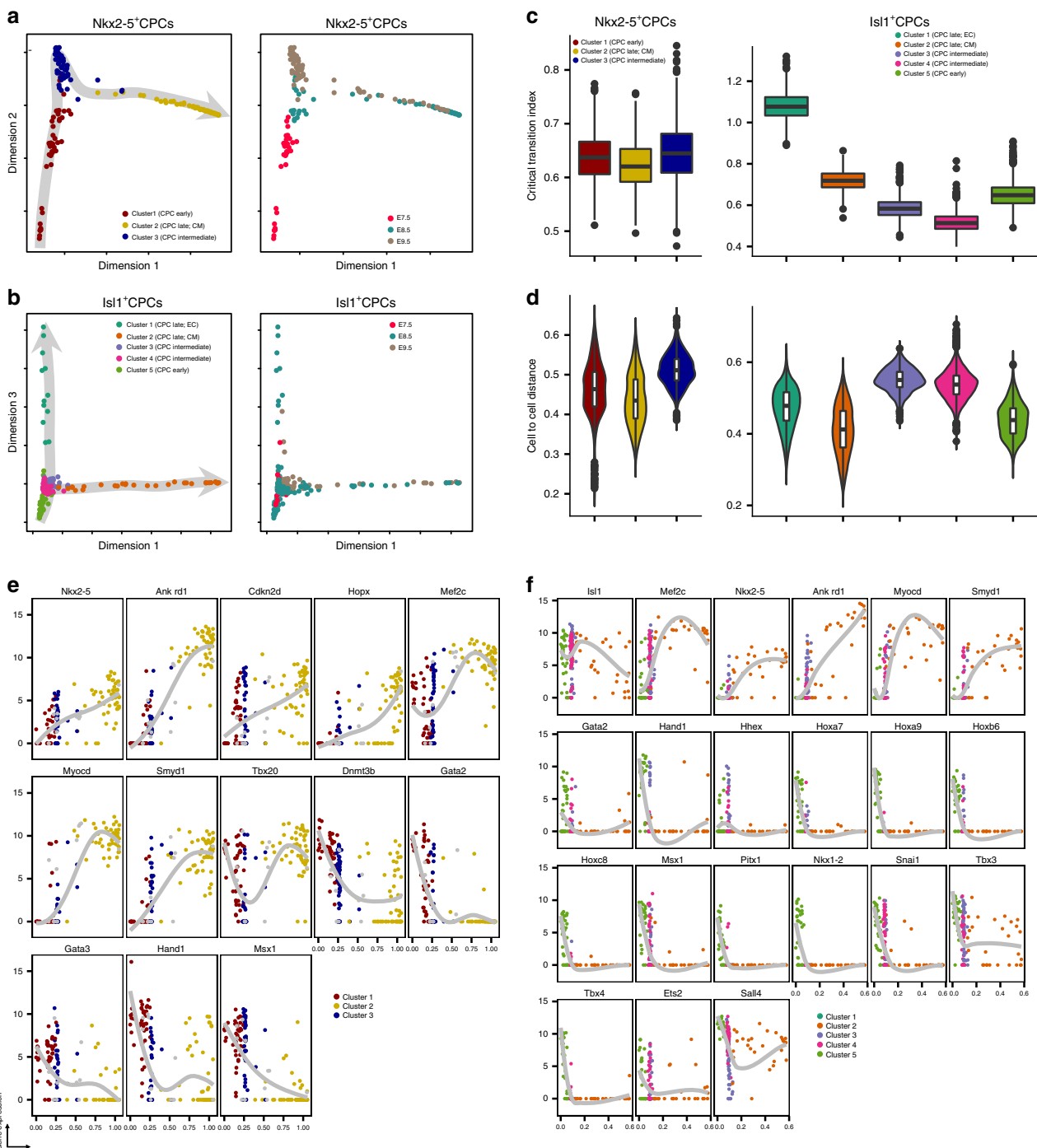

**Fig. 2** Reconstruction of trajectories and transition states of CPCs. **a** t-SNE plots showing diffusion pseudotimes (gray arrows) of Nkx2-5+ and **b** Isl1+ CPCs. Clusters and development stages of individual cells are color-coded as indicated. **c** Boxplots representing the distribution of $I_C(C)$ values from all marker genes for each cluster of Nkx2-5+ (left) and Isl1+ (right) cells. Lower and upper hinges correspond to the first and third quantile (25th and 75th percentile), while whiskers extend from the hinge to the smallest (largest) datum not further than 1.5 times the interquartile range. Outliers are plotted individually. **d** Violin plots showing the distribution of pairwise cell-to-cell distances across each cluster of Nkx2-5+ (left) and Isl1+ (right) cells. Inset boxplots show the median, lower and upper hinges as well as whiskers and outliers as in (**c**). **e, f** Expression levels of different transcription factors and key marker genes on the pseudotime axis in Nkx2-5+ (**e**) and Isl1+ (**f**) cells. Trend lines calculated by Loess regression are indicated in gray. Source data for (**c–f**) are provided in the Source Data file

Isl1$^{nGFP/nGFP}$ embryos and isolated Isl1-GFP+ cells by FACS analysis at E9.5 (Fig. 5a). Projection of *Isl1*-KO single cells on the trajectory of the developing SHF revealed that Isl1-KO cells are stalled/trapped in the previously identified stable attractor state (Fig. 5b). Analysis of G1/S and G2/M cell cycle markers in single cells (Supplementary Fig. 9a–c) suggested reduction of cycling

*Isl1*-knockout cells compared to wild-type cells, although the results did not reach statistical significance ($\chi^2$ test: $p = 0.062$) (Fig. 5c). We concluded that proliferation defects might contribute to the attractor state of *Isl1*-knockout cells but that additional biological processes probably play more important roles. To identify such processes, we performed gene ontology

analysis of deregulated genes in *Isl1*-KO cells. Interestingly, cell differentiation genes were strongly affected by the inactivation of Isl1 (Fig. 5d). Moreover, the GO terms endothelial cell migration and muscle organ development were enriched in WT compared to *Isl1*-knockout cells (Fig. 5d), which is consistent with compromised differentiation of *Isl1*-knockout cells to endothelial cells or cardiomyocytes.

**Nkx2-5 establishes a unidirectional fate for CPCs.** Our pseudotime-based analysis of developmental trajectories revealed one continuous trajectory of Nkx2-5$^+$ CPCs suggesting that Nkx2-5$^+$ cells are exclusively committed to become cardiomyocytes. Although the previous lineage tracing studies strongly indicate that Nkx2-5$^+$ CPCs are multipotent[30], we reasoned that cardiac priming at E8.5 due to continued expression of *Nkx2-5* might overcome smooth muscle identity and induce a stable cardiomyocyte fate. To directly test this hypothesis, we first re-

analyzed published scRNA-seq data of *Nkx2-5* null embryonic hearts at E9.5[19] and found significantly increased numbers of smooth muscle cells raising from 14.5% (138/949 cells) in wild type to 31.2% (39/125 cells) in *Nkx2-5* mutant hearts ($\chi^2$ test: $p <$ 2.37e−6) (Fig. 6a). In a second approach, we specifically expressed *Nkx2-5* and *EGFP* (separated by an IRES) in the Isl1$^+$ lineage using Isl1-Cre to initiate transcription from the *Rosa26* locus (hereafter named Isl1$^+$/Nkx2-5OE)[8]. Isolation of GFP$^+$ cells by FACS from E12.5 embryonic hearts and scRNA-seq (Fig. 6b) revealed that Isl1$^+$/Nkx2-5OE cells align to the Nkx2-5$^+$ trajectory and the cardiomyocyte-like branch of the Isl1$^+$ trajectory (Fig. 6c, d). Importantly, Isl1$^+$/Nkx2-5OE cells did not contain any endothelial cell- or smooth muscle cell-like populations, although by E12.5 Isl1$^+$ cells have given rise to multiple endothelial cells in wild-type conditions (Fig. 6d) indicating that *Nkx2-5* is required and sufficient to resolve the multipotent differentiation capacity of CPCs.

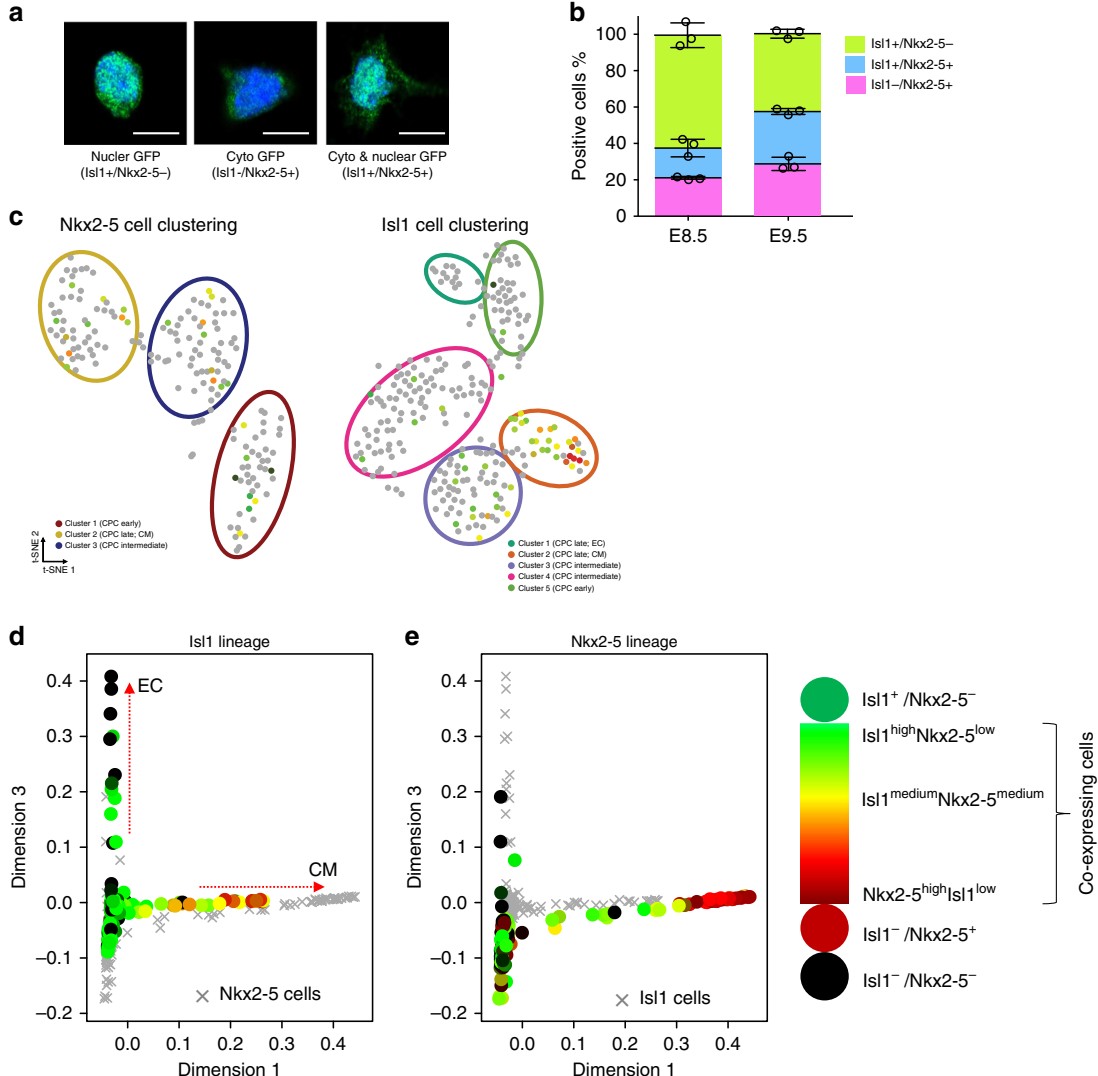

**Fig. 3** Comparison of Isl1$^+$ and Nkx2-5$^+$ cardiac progenitor cells. **a** Confocal images showing nuclear-, cytoplasmic- and co-localization of GFP in CPCs FACS-sorted from Isl1$^{+/nGFP}$/Nkx2-5-emGFP$^+$ embryos. Nuclei were stained with DAPI (blue). **b** Immunofluorescence-based quantification of (**a**). Isl1$^+$Nkx2-5$^-$, Isl1$^+$Nkx2-5$^+$ and Isl1$^-$Nkx2-5$^+$ cells were FACS-sorted from Isl1$^{+/nGFP}$/Nkx2-5-emGFP$^+$ embryos at E8.5 and E9.5. Quantification of different cell populations was achieved by counting all immunostained cells in a multiwell dish. Mean ± s.d. are shown. Circles represent results from different biological replicates [$n = 3$; $\Sigma$ (cell number) of E8.5 = 225, 260, 100; $\Sigma$ (cell number) of E9.5 = 175, 180, 100]. **c** Clustering of Isl1 and Nkx2-5 co-expressing cells in Nkx2-5$^+$ and Isl1$^+$ CPC subpopulations. Cells that are not double-positive are labeled in gray, and clusters are indicated by colored circles. **d**, **e** Plots showing the predicted diffusion pseudotime of Nkx2-5$^+$ cells projected on t-SNE plots of Isl1$^+$ cells, and the expression of Isl1$^+$ (**d**) and Nkx2-5$^+$ (**e**). Expression levels of Isl1 and Nkx2-5 in CPCs are represented by a color spectrum as indicated. EC, endothelial cell. CM, cardiomyocyte

**Single cell chromatin accessibility of Isl1⁺ CPCs.** Profiling of genome-wide chromatin accessibility using bulk preparations of cells will only identify opening of cis-regulatory elements at the population level. Thus, we utilized a single cell ATAC-seq (scATAC-seq) approach[32] and analyzed E8.5 and E9.5 FACS-sorted Isl1⁺ CPCs. The aggregated reads from all individual CPCs closely recapitulated the open regions recognized by bulk ATAC-seq using 2000–50,000 cells (Fig. 7a). After various stringent quality assessments and filtering (Supplementary Fig.11a–d), 67,368 peaks out of 695 sequenced CPCs were analyzed.

The sparse and binary nature of scATAC-seq data poses new computational challenges for data analysis[33]. We employed methods of information retrieval[32] to weight important peaks enabling detection of 5 different subpopulations (Fig. 7b).

Notably, CPCs sampled at E8.5 and 9.5 were evenly distributed between cluster 1, 2, and 3, again arguing for differential, albeit continuous, progression through the developmental program. Cells in cluster 4 and 5 were mainly derived from E9.5, suggesting that they had reached a certain level of maturity. To obtain insights into the biological processes in specific CPC populations, we identified cluster-specific peaks (Supplementary Data 4)[34] and performed gene ontology analysis of genes that were close to (within ± 2.5 kb to TSSs) proximal, cluster-specific peaks. Cluster 1 was enriched for GO terms related to heart development and muscle contraction suggesting advanced cardiomyocyte differentiation, whereas cluster 5 showed enrichment of GO terms related to endothelial cells (Fig. 7c). Consistently, cluster 1 contained the cardiomyocyte genes *Hand1* and *Cacna1c*, whereas

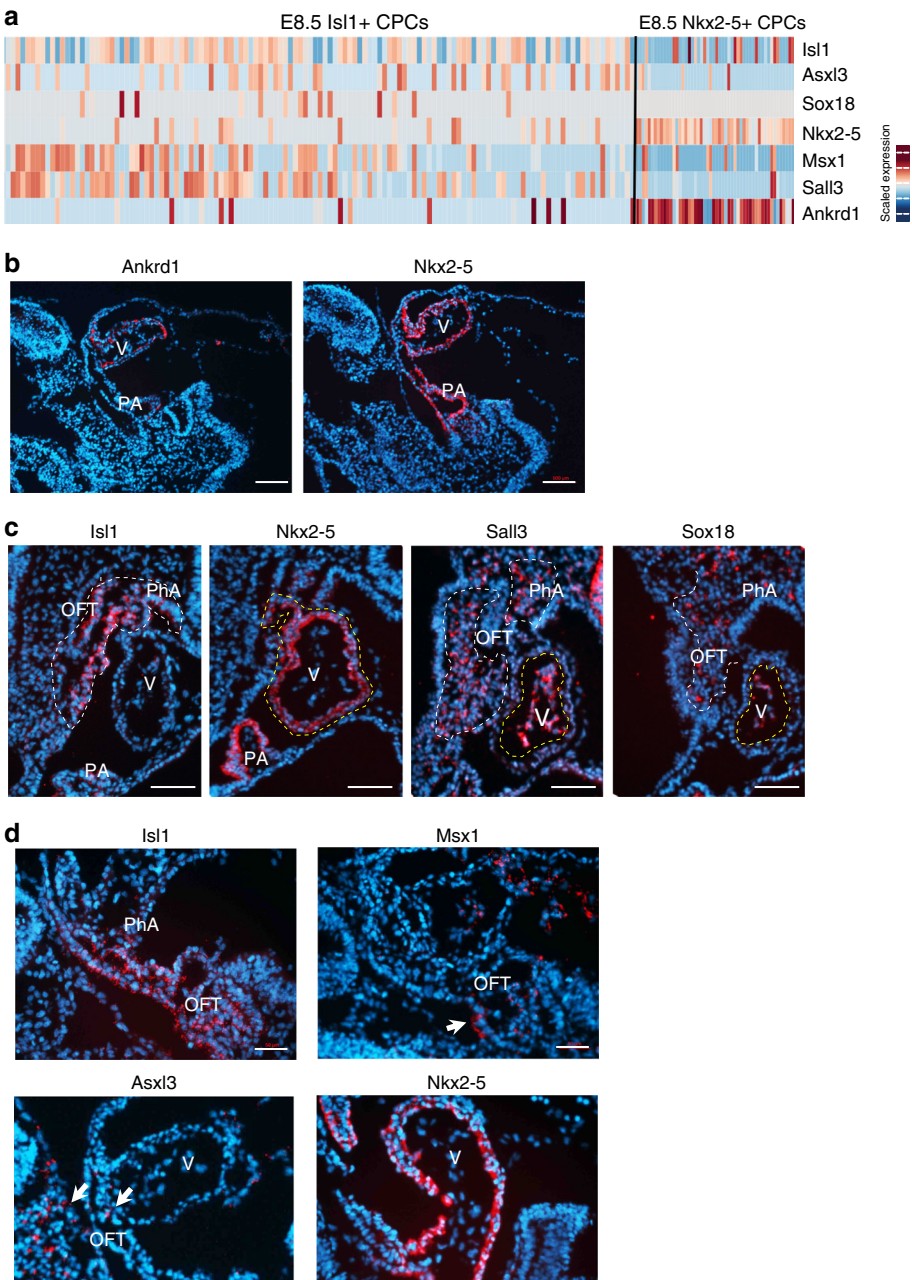

**Fig. 4** Spatial expression pattern of genes identified by scRNA-seq of CPCs. **a** Heatmap showing expression of selected genes in Isl1⁺ and Nkx2-5⁺ CPCs at E8.5. **b–d** In situ hybridization of sections from E8.5 embryos to reveal spatial expression profiles of genes identified by scRNA-seq. Scale bar: 100 μm for (**b**), 50 μm for (**c**, **d**). V: ventricle. PA: primitive atria. PhA: pharyngeal arches. OFT: outflow tract. Arrows indicate positive cells

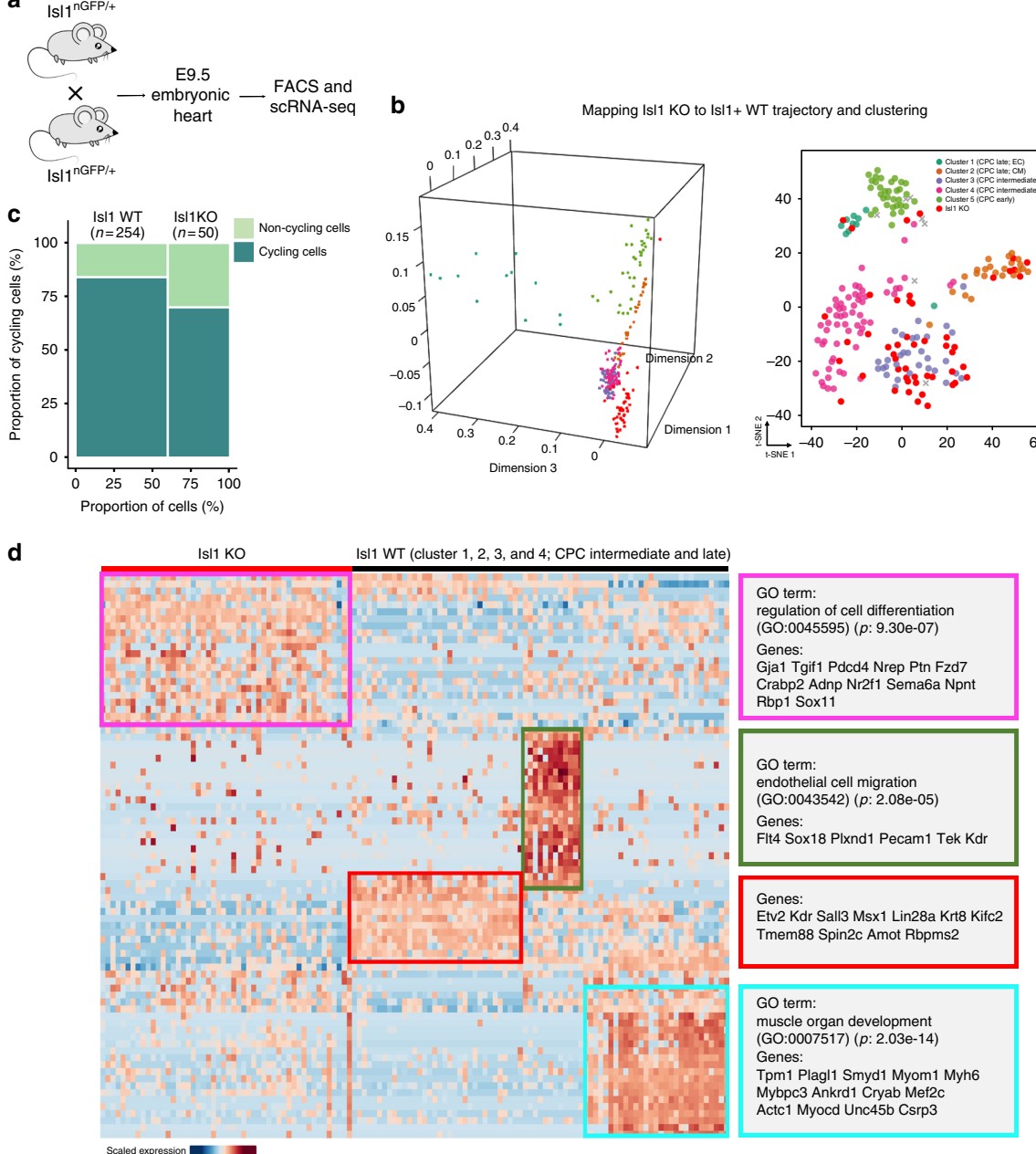

**Fig. 5** Inactivation of Isl1 prevents CPC fate bifurcation. **a** Schematic illustration depicting generation of Isl1 embryos and scRNA-seq. **b** t-SNE plots showing the predicted diffusion pseudotime of Isl1 knockout CPCs projected on Isl1$^+$ cells (left), and clustering with Isl1$^+$ cells (right). **c** Ratios of cycling and non-cycling Isl1 knockout and wild type Isl1$^+$ CPCs. $\chi^2$ test: $p = 0.062$. $n$ indicates cells numbers. **d** Heatmap showing expression of deregulated genes in Isl1$^+$ cells at E8.5 and E9.5 (cluster 1, 2, and 5) isolated from Isl1 knockout and control embryos. Source data are provided in the Source Data file

cluster 5 comprised the endothelial cell genes *Tek* and *Ecm1* (Supplementary Fig. 12a). Cluster 2 was enriched for GO terms related to embryonic morphogenesis genes typical for early progenitor cells preceding bifurcation into cardiomyocyte and endothelial lineages.

**Settlement of the Isl1$^+$ lineage by transcription factors.** ATAC-seq provides an excellent tool to identify transcription factor (TFs) motifs that become accessible due to nucleosome eviction and/or chromatin remodeling[13]. Using chromVar[35], we focused on transcription factor dynamics and variations in motif accessibility taking into account that TF motifs identified by ATAC-seq frequently do not distinguish between related TFs of the same

family usually sharing similar motifs[36]. We ranked cluster-specific TFs and subjected top scorers to t-SNE. In line with the heterogeneity revealed by scRNA-seq analysis, chromVAR identified 5 subpopulations from Isl1$^+$ CPCs (Fig. 8a; Supplementary Fig. 12b). We next aligned the TF motif patterns of individual single cells into a pseudotemporal ordering (Fig. 8b). Based on the gradual and continuous change of motif patterns in each cluster and the annotations of major subpopulations, we split the cells into cardiomyocyte and endothelial trajectories (Fig. 8c, d).

Pseudotemporal ordering indicated TF binding dynamics in different developmental branches, suggesting that a set of TFs cooperatively regulates Isl1$^+$ CPC differentiation: (i) cluster 2 CPCs feature Zeb1, Tcf3/4, and Fox family TFs; (ii) established

cardiac TFs such as Tbx5, Hand 1 and Gata family, together with *Hox* genes are closely associated with cardiomyocyte lineage settlement in cluster 1; (iii) the induction of *Sox* gene family in transition CPCs (cluster 3) seems to skew the cell fate to the endothelial lineage, which (iv) eventually is characterized by enrichment of Gata and Sox TF binding (Fig. 8e). Correlation of RNA expression and chromatin accessibility in individual single cells revealed two characteristic patterns of ATAC:RNA pairs: (i) RNA expression of TFs directly matches accessibility of corresponding TF bindings sites as exemplified for *Hox* and *Gata* families in the cardiac branch suggesting that members of both TF families actively regulate respective target genes at this developmental stage; (ii) RNA expression of TFs precedes accessibility of corresponding TF bindings sites. This scenario was apparent for *Sox* and *Gata* families in the endothelial branch, suggesting that additional epigenetic regulatory mechanisms have to occur before TFs take action (Fig. 8f).

**Isl1 shapes chromatin accessibility in CPCs.** To analyze how the lack of Isl1 expression affects chromatin accessibility, we performed bulk ATAC-seq of Isl1$^+$ mutant CPCs at E9.5. Since our scRNA-seq analysis at E9.5 indicated that Isl1-KO cells are trapped in the previously identified stable attractor state, we compared ATAC-seq data from Isl1$^+$ mutant CPCs to Isl1$^+$ WT CPCs from E8.5 and E9.5, which were separated into Isl1

$^+$/CD31$^-$ CPCs (CM-trajectory) and Isl1$^+$/CD31$^+$ CPCs (EC-trajectory) (Supplementary Fig. 13a). For each condition and time-point, at least two biological replicates were used (Supplementary Data 5, 6; Supplementary Fig. 13b, c). Inactivation of the *Isl1* gene resulted in barely any changes compared to Isl1$^+$/CD31$^-$ CPCs (CM-trajectory) either at E8.5 or E9.5 (Fig. 9a, b). However, loss of *Isl1* led to more robust closing than opening peaks compared to Isl1$^+$/CD31$^+$ CPCs (EC-trajectory) at both E8.5 and E9.5, suggesting that *Isl1* is required to leave the attractor state characterized by a more open chromatin organization (Fig. 9a, b). In a nutshell, these results indicate that *Isl1* mutant CPCs exhibit an epigenomic profile that strongly resembles developing cardiomyocytes and differs from endothelial cells.

Next, we focused on differences in chromatin accessibility between Isl1$^+$/CD31$^+$ CPCs and *Isl1* mutant CPCs. Annotation of opening and closing peaks by GREAT analysis[37] indicated that 13.5% of differential peaks (1120) located in proximal regions whereas 87.5% (7260) were present in distal regions (> ± 5 kb to the TSS sites) (Fig. 9c; Supplementary Fig. 13c). To investigate whether changes in chromatin accessibility in proximal regions correlated with differential gene expression, we generated additionally transcriptional profiles of biological replicates at each corresponding developmental stage by bulk RNA-seq using the SMART-seq2 method[38]. We paired 126 elements that were more accessible after inactivation of Isl1 and 636 elements more

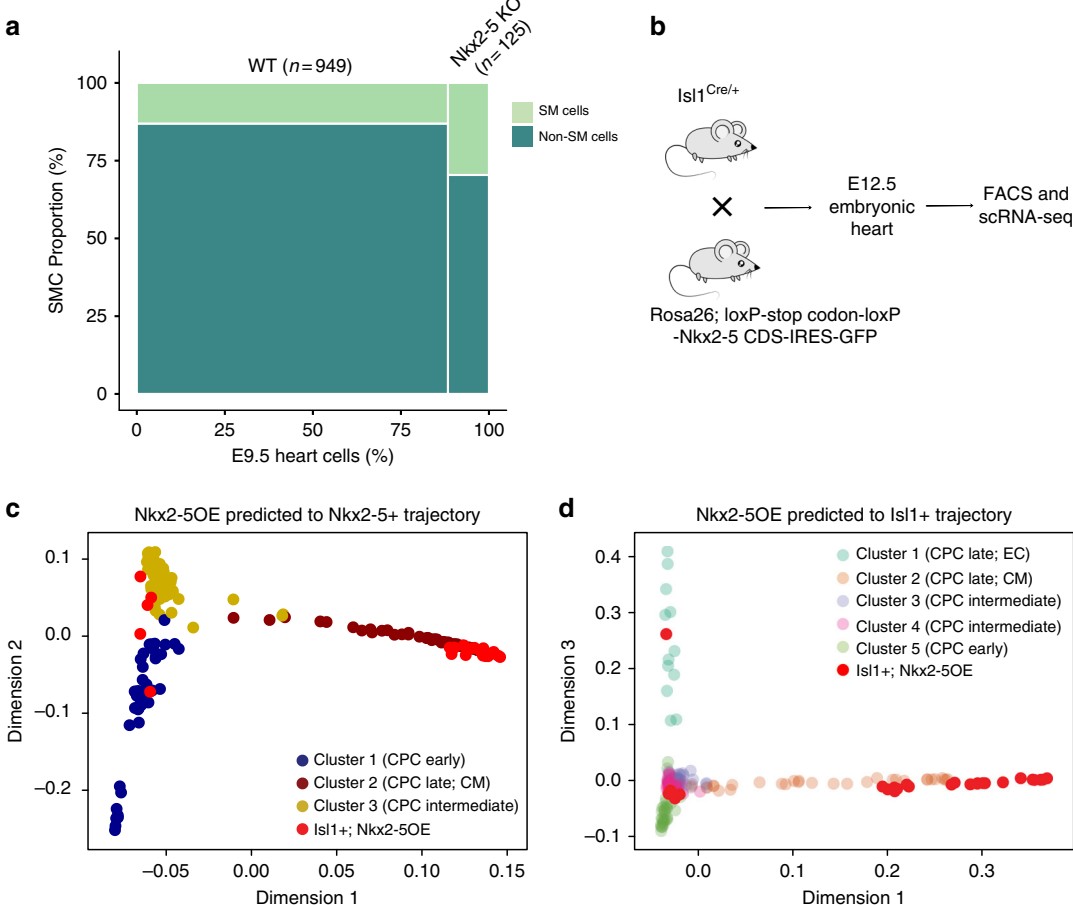

**Fig. 6** Nkx2-5 institutes a unidirectional fate in CPCs to cardiomyocytes. **a** Re-analysis of published data showing the ratio of smooth muscle cells in embryonic hearts of wild type and Nkx2-5 knockout embryos at E9.5. Smooth muscle cells are scored by low expression of *Nkx2-5* (LogTPM < 1, null expression) and high expression of smooth muscle cell genes (*Tagln, Cnn1, Acta2, Cald1, Mylk, Hexim1,* and *Smtnl2* moderate to high (LogTPM > 2) for at least 5 of these 7 genes). $\chi^2$ test: $p < 2.37e{-}6$. *n* indicates cells numbers. **b** Schematic illustration of forced expression of Nkx2-5 in Isl1$^+$ cells and scRNA-seq. **c** Predicted diffusion pseudotime of Isl1$^+$/Nkx2-5OE cells projected on t-SNE plots of Nkx2-5$^+$ and Isl1$^+$ **d** CPCs

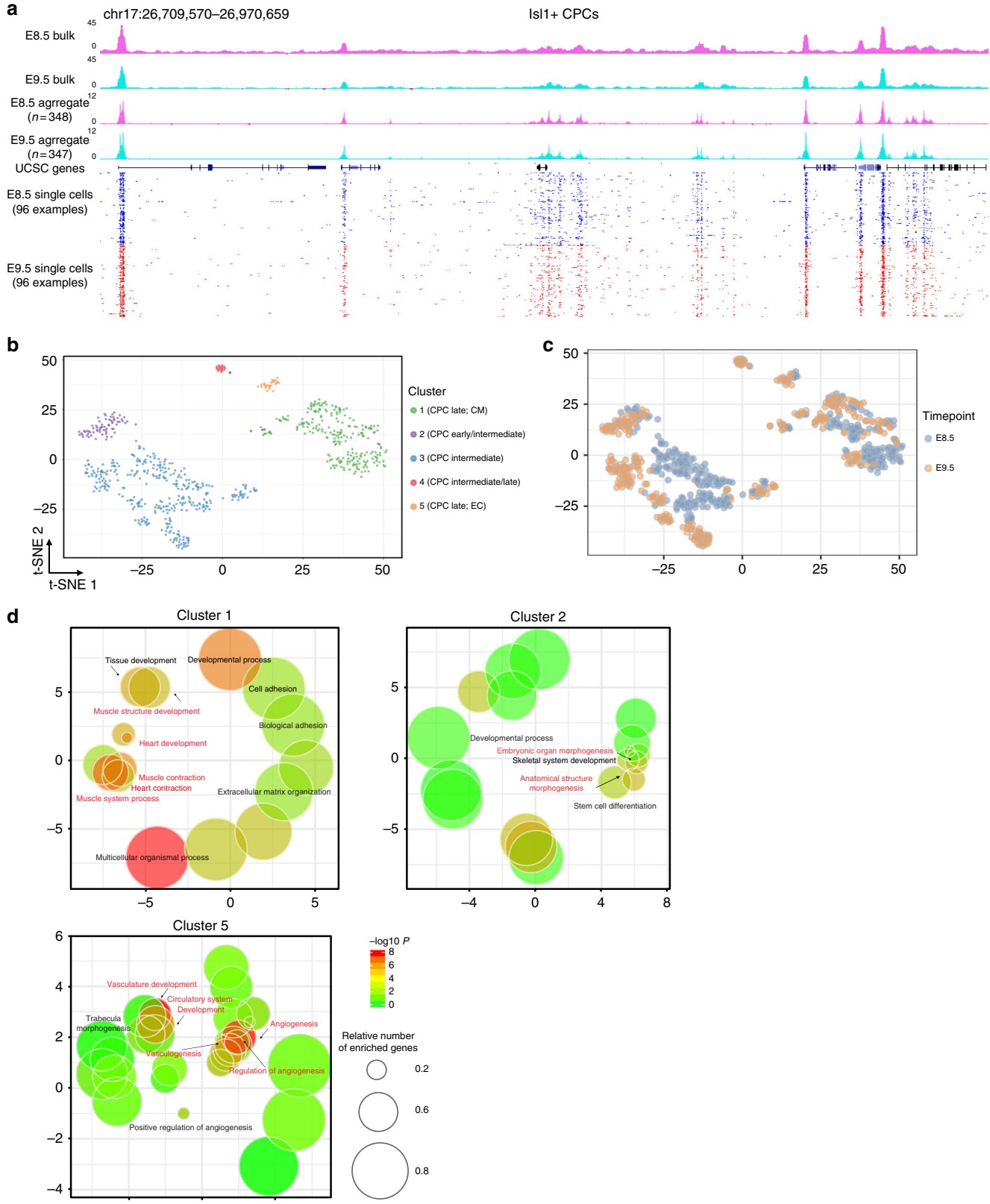

**Fig. 7** Single cell chromatin accessibility profiles of Isl1+ CPCs. **a** Representative genomic region showing ATAC-seq tracks of single, aggregate and bulk cells. **b, c** t-SNE visualization of individual Nkx2-5+ and Isl1+ CPCs to identify subpopulations based on chromatin accessibility. Colors denote corresponding clusters (**b**), and (**c**) development stages. **d** Gene ontology (GO) enrichment analyses of scATAC-seq clusters 1, 2, 5 of Isl1+ CPCs. Each bubble represents one of the top enriched GO terms. The relevant GO terms ($p < 0.05$, calculated from the hypergeometric distribution) are highlighted

accessible in Isl1+/CD31+ CPCs (EC-trajectory) with corresponding gene expression. On average, promoters that gain chromatin accessibility displayed significant upregulation of gene expression, whereas gene loci losing chromatin accessibility down-regulated expression ($p < 0.001$, Student's $t$-test) (Fig. 9d), revealing a clear correlation between chromatin accessibility and

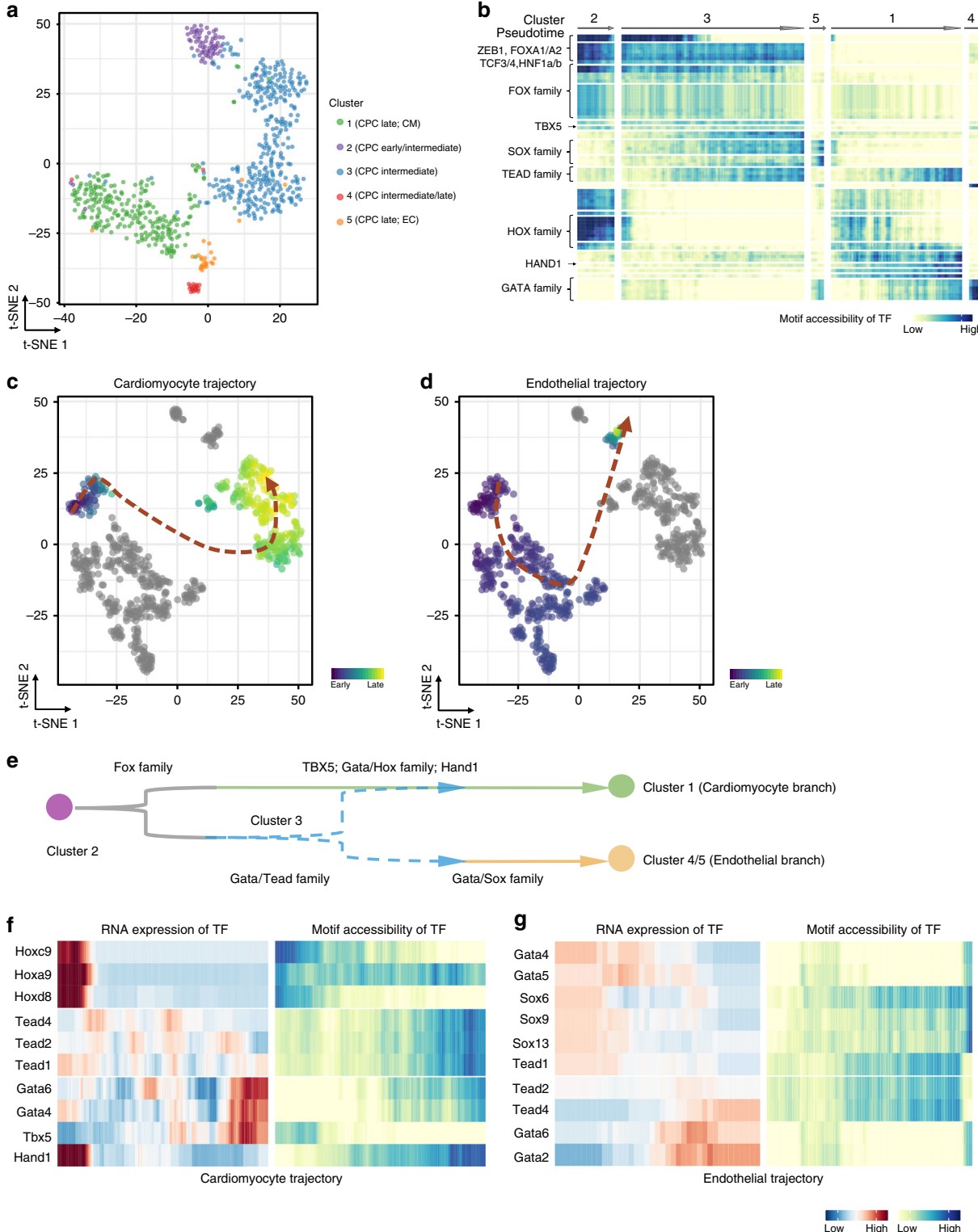

**Fig. 8** Chromatin accessibility of transcription factor binding sites. **a** t-SNE showing clustering of $Z$-scores of TF motif accessibility. Colors denote the same clusters as Fig. 7b. **b** Heatmap showing smoothened $Z$-scores of TF motif accessibility across defined clusters. Source data are provided in the Source Data file. **c**, **d** t-SNE visualization of highlighted single-cells progressing through the inferred (**c**) cardiomyocyte, (**d**) endothelial developmental trajectory (red dashed lines). Cells used for inference are colored by $Z$-scores of TF motif accessibility. All other cells are shown in gray. **e** Inferred model showing TF dynamics during Isl1+ CPC developmental bifurcation. **f**, **g** Smoothened heatmap showing dynamic RNA expression and motif accessibility of indicated TFs during cardiomyocyte (**f**), endothelial (**d**) pseudotime trajectories for gene-motif pairs (RNA:ATAC pairs). EC, endothelial cell. CM, cardiomyocyte

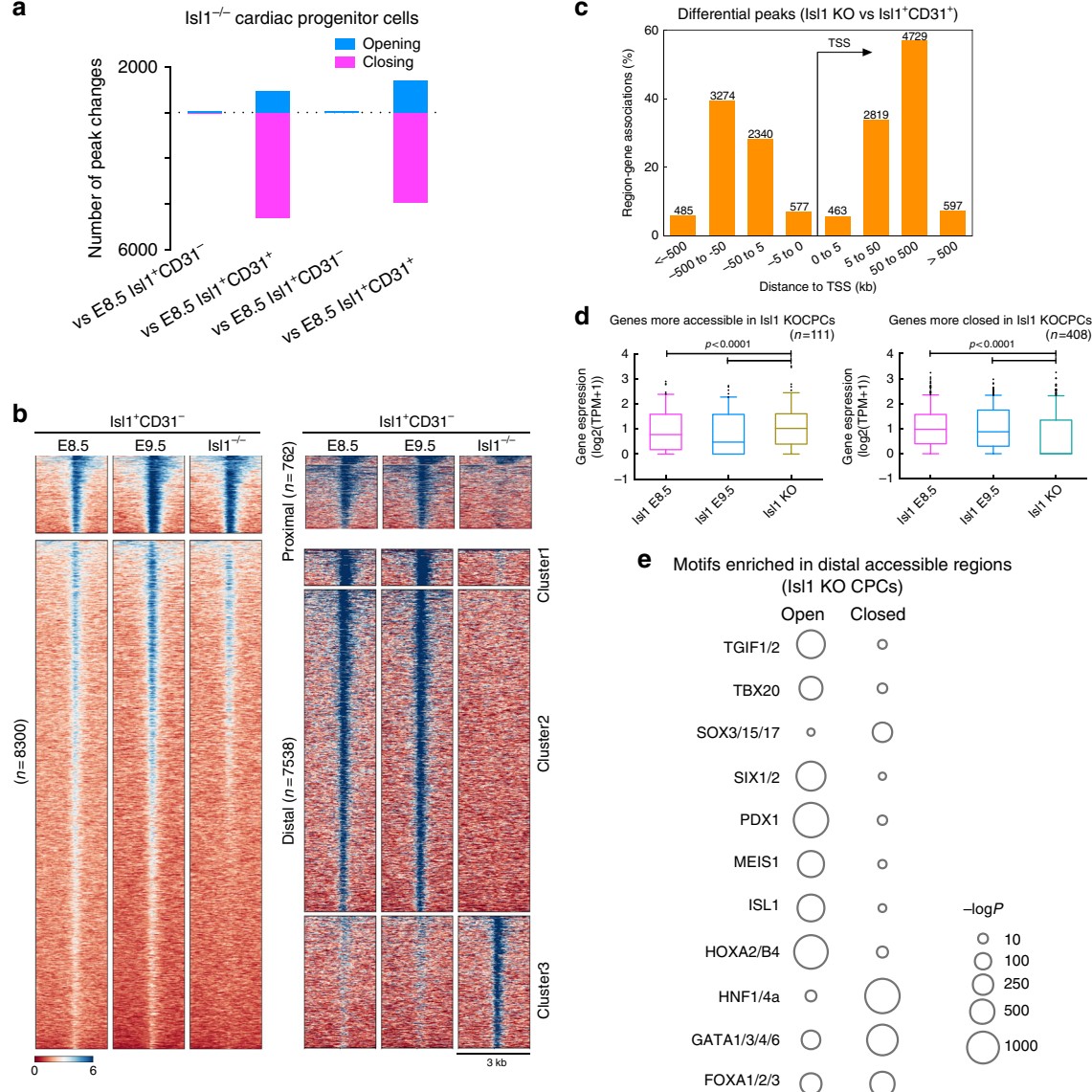

**Fig. 9** Chromatin accessibility in CPCs is shaped by Isl1. **a** Number of differential chromatin accessibility peaks (log2(FC) > 2, false discovery rate [FDR] < 0.05). **b** Genome-wide distribution of differential open chromatin peaks grouped by $K$-means (left), and by distance to promoter and $K$-means (right). Each row represents one differential peak, normalized to sequencing depth, in sequential comparisons (log2[FC] > 2, FDR < 0.05). **c** Number of differential peaks and their distance to the nearest promoters. **d** Boxplots of mRNA expression levels in E8.5 and 9.5 Isl1+ CPCs, and of genes that are more accessible (left) or more closed (right) in Isl1 KO cells. Box lines show the median, 25th and 75th percentiles; whiskers represent 5th and 95th percentiles; dots represent outlier data points. $p$-values were calculated using Student's $t$-test. $n$ indicates the genes numbers. **e** Bubble chart showing the enrichment of transcription factor motifs in differential peaks ($p$-values were calculated from the hypergeometric distribution)

nearby gene expression. Based on the enrichment of H3K27ac and H3K4me1 (Supplementary Fig. 13d), we reasoned that the distal peaks most likely represent enhancer regions[39].

Assessment of transcription factor motifs present in either opening or closing peaks using the motif analysis package HOMER[40] revealed enrichment of binding motifs for Tbx20, Meis1 and Hox-family members in *Isl1*-dependent opening peaks while binding sites for Sox-, HNF- and GATA-family TFs were enriched at peaks that disappeared after loss of *Isl1* (Fig. 9e). We concluded that *Isl1* acts together with *Tbx* genes to guide cardiac progenitor cell fate decisions but prevents binding of Sox and HNF factors for endothelial cell fate termination.

**The chromatin accessibility landscape of Nkx2-5+ CPCs.** Bulk ATAC-seq analysis of FACS-sorted Nkx2-5+ CPCs sampled at

E7.5, E8.5, and E9.5 revealed major genome-wide changes of chromatin accessibility between E7.5 and E8.5, while E9.5 CPCs showed only minor differences compared to E8.5 Nkx2-5+ CPCs suggesting that by E8.5 Nkx2-5+ CPCs have already opened or closed most genomic loci required for further development towards cardiomyocytes (Fig. 10a; Supplementary Fig. 14a; Supplementary Data 7). A more refined analysis of the genome-wide distribution of differential peaks by $K$-means clustering identified regions in cluster 1 that were closed at E7.5 but open at E8.5 (Fig. 10b). Genes associated with the opening regions in cluster 1 were intimately involved in sarcomere and contractile fiber formation (Supplementary Fig. 14b). Interestingly, this group of genes was characterized by peaks located in proximal regions (Fig. 10c), which comprised only 6.7% of all peaks. The majority of changes in chromatin accessibility in cluster 1 was located in

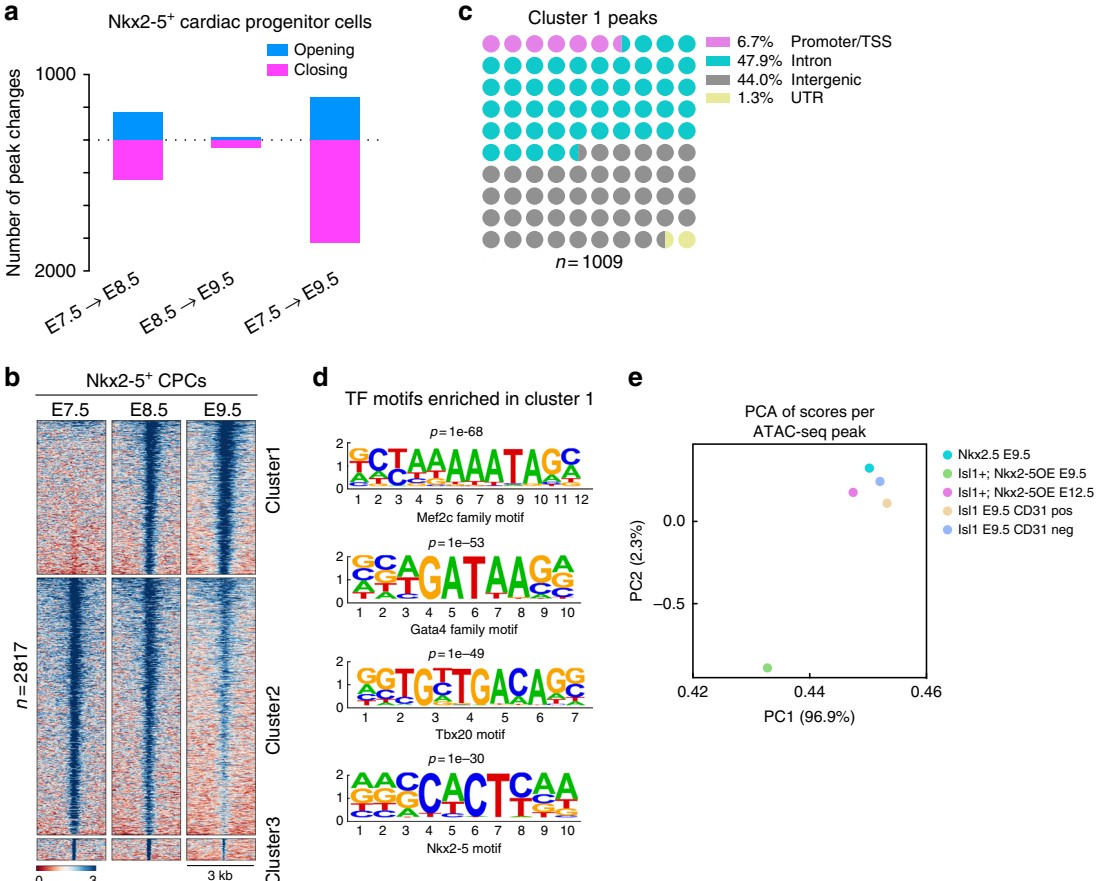

**Fig. 10** Bulk ATAC-seq analysis of Nkx2-5[+] CPCs. **a** Number of differential chromatin accessibility peaks (log2(FC) > 2, false discovery rate [FDR] < 0.05). **b** Genome-wide distribution of differential open chromatin peaks grouped by *K*-means. Each row represents one differential peak, normalized to sequencing depth, in sequential comparisons (log2[FC] > 2, FDR < 0.05). **c** Distribution of genomic features of differential regulatory elements. **d** Enrichment of known transcription factor motifs in Isl1[+]/Nkx2-5OE differential peaks. The height of the letters represents the frequency of each base in the cognate motif. **e** Principle component analysis (PCA) showing chromatin accessibility features among indicated samples by reducing dimensionality

distal regions, which were enriched in TF motifs for Mef2c, Gata4, Tbx20, and Nkx2-5 (Fig. 10d). Taken together our newly established chromatin accessibility atlas of Nkx2-5[+] CPCs identified a set of TFs that seems to act together with Nkx2-5 for orchestration of Nkx2-5[+] cell maturation.

In contrast to the inactivation of *Isl1*, which preferentially erased accessible chromatin sites (Fig. 9a), forced expression of *Nkx2-5* increased accessible chromatin sites at multiple loci compared with either Nkx2-5[+] or Isl1[+] CPCs at E9.5 and E12.5 (Supplementary Fig. 14c, d; Supplementary Data 7). The profound opening of chromatin at E9.5 after overexpression of *Nkx2-5*, which is particularly evident in cluster 1 and 4 (Supplementary Fig. 14e), is a striking example for the ability of single TFs to alter chromatin structure. Although comparatively minor, we also observed enhanced chromatin closing in clusters 2 and 3 at E9.5, which is probably initiated by secondary events (Supplementary Fig. 14f). Principle component analysis (PCA) of all unified peaks revealed that *Nkx2-5* overexpression induces a distinct profile of accessible chromatin sites at E9.5, which is different compared to what is seen in other CPCs (Fig. 10e). However, we noted that de novo chromatin opening occurred only transiently but was not sustained upon CPC differentiation, since E12.5 *Nkx2-5* overexpressing cells compared to E9.5, grouped closer to E9.5 Isl1[+] and Nkx2-5[+] CPCs in the PCA analysis (Fig. 10e). We hypothesized that chromatin opening evoked by *Nkx2-5* overexpression was overcome by cellular events set in motion by *Nkx2-5* during differentiation. Analysis of Nkx2-

5 OE-dependent differential peaks revealed enrichment of Gata- and Mef2c-motifs, consistent with cardiomyocyte fate determination by *Nkx2-5* (Supplementary Fig. 14f).

## Discussion

Our scRNA-seq analysis provides a rich data source for the discovery of genes that might play a role in heart development. For example, we found that posterior *Hox* genes are temporarily expressed in early-stage Isl1[+] cells. Anterior *Hox* genes (*Hoxa1*, *Hoxb1*, and *Hoxa3*) are involved in cardiac development[41–43] but expression of posterior *Hox* genes had not been detected in CPCs so far. We speculate that posterior Hox family TFs might contribute to patterning of the heart, which is supported the presence of cardiac defects in *Hox A/B* cluster compound mutants[44]. Our scATAC-seq data also revealed enriched binding of *Hox* TFs in Isl1[+] CPCs of the cardiomyocyte branch, further supporting the potential function of *Hox* TFs for heart development and cardiomyocyte differentiation. Surprisingly, our results indicate that *Isl1* and *Nkx2-5* co-expression does not drive the formation of a distinct cluster characterized by a specific gene expression profile. Instead, *Nkx2-5* expression serves as a marker for late-stage CPCs that already express cardiac genes and are about to abandon *Isl1* expression.

Numerous studies have demonstrated that cells within apparently homogenous populations differ considerably, which might have many reasons ranging from stochastic fluctuations in gene

expression patterns, different stages of cell division, execution of subspecialized tasks to differential priming for future cellular decisions[26,27,45,46]. Cellular heterogeneity is particularly high during embryonic development when multipotent cells have to undergo a series of decisions to acquire a differentiated phenotype[19,20,47]. Notwithstanding, it was surprising that the cardiogenic Isl1+ and Nkx2-5+ CPCs own a staggering degree of heterogeneity. Several reasons might account for this finding: (i) due to the massive chromatin and transcriptional changes that are required, it might cost individual cells different time to pass through transition states[25] or switch between different attractor states[26], (ii) cells within a cluster defined by scRNA-seq differ in regard to open or closed loci, which will change responsiveness to external cues, causing secondary heterogeneity, (iii) developmental decisions are often driven by gradients of signaling molecules[22], which necessarily cause differential responses and heterogeneity, (iv) the parallel existence of heterogeneous clusters of cells in a specific lineage creates more flexibility and adaptability that enhances the ability to cope with turbulences during inductive events, thus increasing the robustness of developmental processes.

Lineage tracing and clonal analyses have demonstrated that Nkx2-5+ CPCs contribute to cardiac endothelium and smooth muscle cells[30]. However, we did not detect Nkx2-5 expression outside of the cardiomyocyte lineage during differentiation, which seems to be in conflict with lineage tracing studies but is consistent with prior studies allocating Nkx2-5 primarily to cardiomyocytes[20]. It is important to remember that tracing or actual reporter-based gene activity approaches address different questions. In our study, we exclusively focused on Nkx2-5 expressing cells but not on their derivatives, which excludes Nkx2-5-derived cells that have terminated Nkx2-5 expression. We reason that Nkx2-5 expression is essential to maintain the ability of multipotent progenitor cells to differentiate into cardiomyocytes but that Nkx2-5 expression is quickly terminated in cells acquiring a stable endocardial or smooth muscle cell fate thereby escaping Nkx2-5-emGFP based FACS-sorting. Interestingly, scRNA-Seq analysis indicated that Nkx2-5+ cells at E8.5 express cardiac markers such as cTNT but also α-SMA as well as several other smooth muscle markers such as Caldesmon, Tagln, and Cnn1 (Fig. 1f, g; Supplementary Fig. 10). The co-expression of cardiomyocyte and smooth muscle cell markers might suggest the ability of Nkx2-5+ cells to differentiate into cardiomyocytes and smooth muscle cells but alternatively might reflect the well-known expression of smooth muscle genes in immature cardiomyocytes[48]. We think that the loss of a bipotent fate of Isl1+ cells and the acquisition of a unipotent cardiomyocyte fate after forced expression of Nkx2-5 clearly argue for a decisive role of Nkx2-5 in cardiomyogenic differentiation. In agreement with this hypothesis, expression of cardiomyocyte genes such as Myl7[31,49] increased together with Nkx2-5 expression at early development stages.

So far dynamic changes in the genome-wide chromatin landscape have not been systematically investigated during early heart development, although chromatin remodeling has been linked to heart development and the BAF chromatin-remodeling complex was identified as a crucial factor[49,50]. Our scATAC-seq data demonstrates that profiling of chromatin accessibility in single cells is a powerful tool to uncover cellular heterogeneity, equal or even superior to scRNA-seq. In fact, scATAC-seq detected 5 subpopulations of Isl1+ CPCs at E8.5 and E9.5 compared to 3 subpopulations that we identified by scRNA-seq at this stage. We assume that these differences are not only caused by technical issues but are biologically meaningful. Before genes are expressed, the corresponding loci have to open and enter a euchromatic state, which will be visible by ATAC-seq but not by RNA-seq.

Furthermore, individual TFs might bind to cognate motifs but this might be not sufficient to initiate robust transcription, which will create mismatches between ATAC-seq and RNA-seq data. Of course, such conditions do not apply to all genes, which became evident when we paired scRNA expression with single cell chromatin accessibility. We identified numerous matching ATAC:RNA pairs, which is probably the rule for stable cellular conditions, such as in terminally differentiated cells that do not experience major phenotypical changes. Heterogeneity at the chromatin level in respect to open and closed loci might represent a distinct biological advantage even under ground-state physiological conditions. Differential chromatin accessibility among individual cells will greatly increase flexibility for timely cellular responses within a population of cells by creating permissive or repressive states for transcription.

The integration of different regulatory layers into a comprehensive model that explains different developmental decisions during heart development will be a major challenge for the future. The complex network of stage-specific cis-regulatory elements and the single cell transcription profiles revealed in our study provides part of the essential groundwork to move in this direction.

## Methods

**Mouse work and sampling of single cells.** All animal experiments were performed in accordance with German animal protection laws and EU ethical guidelines (Directive 2010/63/EU) and were approved by the local governmental animal protection authority at the Regierungspräsidium Darmstadt, Germany. The transgenic mouse lines used in this study have been described previously[7,8]. C57BL/6 mouse embryos were dissected at E7.5, E8.5, E9.5 or E12.5. At E8.5, E9.5, and E12.5 we used dissected hearts instead of the whole embryo (at E7.5), to avoid contamination of non-cardiogenic cells that might be marked by Isl1 or Nkx2-5 expression. In total, 403 embryonic hearts were harvested for scRNA-seq, scATAC-seq, bulk RNA-seq, and bulk ATAC-seq experiments. Each embryo was accurately staged based on the number of somites to allow precise matching of different developmental stages. Embryonic hearts were isolated under the dissection microscope and digested into single cells suspensions with 0.25% trypsin-EDTA. After washing with PBS, cells were stained with DAPI to check for viability and sorted using the GFP channel of the BD FACSAria II instrument. To obtain Isl1 nGFP/nGFP or Isl1+/Nkx2-5OE cells, Isl1nGFP/+ mice or Isl1-Cre and Rosa26Nxk2-5-IRES-GFP mice were mated and embryos were recovered at indicated time points, and genotyping was achieved by PCR using non-heart tissue of the same embryos as described[8]. To get Isl1+/nGFP;Nkx2-5emGFP+ embryos, Isl1+/nGFP and Nkx2-5emGFP transgenic mice were mated and embryos were isolated at E8.5 and E9.5, and genotyping was performed as well[8], and Isl1+/nGFP/Nkx2-5emGFP+ (double positive) embryos were retained for FACS-sorting. Following PCR primers were used for genotyping: Isl1nGFP/+ forward: 5′-CTC TTG ATT CCC ACT TTG TGG TTC-3′; Isl1nGFP/+ reverse: 5′-TCA GTA AGC TAT GGG TTA GAG-3′; Isl1-Cre forward: 5′-ACT ATT TGC CAC CTA GCC ACA GCA-3′; Isl1-Cre reverse: 5′-AAT TCA CAC CAA ACA TGC AAG CTG-3′; Nkx2-5emGFP forward: 5′-GAC GTG ACC CTG TTC ATC AG-3′; Nkx2-5emGFP reverse: 5′-GTT TCTT GGG GAC GAA AG-3′; Rosa26Nxk2-5-IRES-GFP forward: 5′-AAA GTC GCT CTG AGT TGT TAT-3′; Rosa26Nxk2-5-IRES-GFP reverse: 5′-GGA GCG GGA GAA ATG GAT ATG-3′.

**FACS-sorting with antibody staining.** To harvest Isl1+CD31- and Isl1+CD31+ CPCs, the dissociated cells as described above were stained with anti-mouse CD31 PE-conjugated antibody (BD Pharmingen Cat# 553373) with the concentration of 1ug/ml on ice for 30 min in the presence of 1% sodium azide, washed with 1 × PBS extensively, and sorted using the GFP and PE channels of the BD FACSAria II instrument.

**Antibody staining and in situ RNA hybridization.** The embryos at indicated developmental stages were harvested and fixed with 4% PFA. The fixed tissue was equilibrated in 10% and 30% sucrose/PBS sequentially and frozen on dry ice. Sections of 10 μm were used for immunofluorescence staining followed the standard protocol. The following antibodies with indicated concentration were used: anti-Nkx2-5 (ThermoScientific Cat# PA5-49431, 1:1,000) and anti-Isl1 (DSHB 39.4D5, 1:100). Purified Isl1+/GFPNkx2-5-emGFP CPCs were stained using anti-GFP antibody (ThermoFisher Scientific Cat# A11120, 1:2,000). The florescent RNA hybridization was performed using ViewRNA ISH Assays (ThermoFisher Scientific Cat# QVT0013) according to the manufacturer's instructions.

**Single-cell RNA sequencing library preparation.** Single-cell capture, lysis, reverse transcription, and pre-amplification were done using C1 chips (#100–5763, 10–17 µm) in the C1 single-cell Auto Prep System (Fluidigm) or the ICELL8™ Single-Cell System (Wafergen) following the manufacturer's protocols. Libraries were sequenced using an Illumina NextSeq 500 system.

**Single cell ATAC-seq and raw data processing.** Mouse CPCs were isolated as described above, and 50,000 cells were cryopreserved in 90% FBS/10% DMSO until further usage. The scATAC-seq experiments were performed as described previously[32]. Isolated cells were incubated in 50 µl tagmentation mix (33 mM Tris-acetate, pH 7.8, 66 mM Potassium acetate, 10 mM magnesium acetate, 16% dimethylformamide (DMF), 0.01% Digitonin and 2.5 ul Nextera Tn5) at 37 °C with 800 rpm for 30 min before adding 50 µl tagmentation stop buffer (10 mM Tris, pH 8.0, 20 mM EDTA) on ice for 5 min. Tagmented single cells were sorted into 384-well plate containing 4 µl lysis buffer (50 mM Tris, pH 8.0, 50 mM NaCl, 20 ug/ml Proteinase K, 0.2% SDS, 10 uM Nextera index primer mix) using a BD-INFLUX sorter.

After sorting, the plate was briefly centrifuged and incubated at 65 °C for 30 min. Then, 4 µl 10% tween-20, 2 µl H2O and 10 µl NEBNext® High-Fidelity 2 × PCR Master Mix were added to each well sequentially. Libraries were amplified with 72 °C 5 min, 98 °C 5 min, [98 °C 10 s, 63 °C 30 s, 72 °C 20 s] × 18. Finally, all reactions were pooled together and purified with a PCR minElute purification column (Qiagen). Libraries were sequenced with a Hiseq2000 machine after size selection.

Reads were trimmed and mapped to the reference mouse genome (UCSC mm10) using hisat2[51]. Reads with mapping quality less than 30 were removed by samtools (-q 30 flag) and deduplicated. All reads from single cells were merged together using samtools, and the merged BAM file was deduplicated again. Peak calling was performed on the merged and deduplicated BAM file by MACS2 resulting in union peaks[52]. A count matrix over the union of peaks was generated by counting the number of reads from individual cells that overlap the union peaks using coverageBed from the bedTools suite[53].

**Bulk ATAC-seq library preparation and sequencing.** In all, 2000–20,000 GFP+ CPCs were FCAS-purified and used for ATAC-seq. The ATAC-seq libraries were prepared as previously described[13]. 2 × 50 paired-end sequencing was performed on Illumina NextSeq500 to achieve on average of 35.08 ± 12.53 million reads per sample (Mean ± s.d.) (Supplementary Data 5).

**Bulk RNA-seq.** A total of 5000–20,000 CPC cells were sampled using the same protocol as described above for scRNA-seq. Bulk RNA-seq libraries were prepared using Smart-seq2 according to the manufacturer's protocol (Cat#634889, Clontech), and sequenced using the Illumina NextSeq500 instrument. Raw reads were processed using the same method as for scRNA-seq. Quantification and identification of differentially expressed genes were carried out using DEseq2[54].

**Single-cell RNA-seq data analysis.** Low-quality bases were trimmed off the raw sequencing reads using Reaper with a minimum median quality of 53 in a window of 20 bases, omitting the first 50 bases of the read. Additionally, the -dust-suffix 20/AT option was used to trim remaining polyA or polyT stretches at the end of reads as well as stretches of B (a special Illumina Quality Score indicating non-trustworthy bases) with the –bcq-late option. The STAR alignment tool was used with default parameters to map trimmed reads to the mouse genome (version mm10) and transcriptome (--quantMode TranscriptomeSAM, together with the Gencode annotation in version vM10). Mapping quality and statistics was assessed using Qualimap in rnaseq mode, setting the protocol to strand-specific-forward and using the same Gencode annotation. The Qualimap output was used later for single-cell filtering (see below). RSEM was used with gene annotations from Gencode vM10 as well as a single-cell prior to assign reads to genes and extract gene-centered counts.

A SingleCellExpression-Set object (SCESet, R package scater) was created in R from all available metadata, cell-quality data, gene annotations, and the gene-centered count table. For each platform (Fluidigm C1 (C1) or Wafergen (WG)), an initial cell-quality map was generated with t-SNE (R package Rtsne) by grouping cells with similar quality metrics together (Supplementary Fig. 1). The (per-cell) quality metrics used as input were: number of features (genes) detected with at least 10 counts, the percentage of gene dropouts, the number of alignments, the number of alignments to exons, introns and intergenic regions, the number of secondary alignments, the expression of Rplp0 (also known as 36B4) as housekeeping gene, the percentage of read counts to mitochondrial genes, as well as the percentage of genes detected.

To define cells as low-quality, we formulated and evaluated five criteria for each cell: The percentage of counts to mitochondrial genes is 1.5 median-absolute-deviations (MADs) above the median, the number of detected features is 2 MADs above or below the median, the percentage of gene dropouts is 2 MADs above the median, the Rplp0 expression is 2 MADs below the median and the percentage of genes is 1.5 MADs above or below the median. Cells failing more than one criterion were considered low-quality and excluded from further analysis. See (Supplementary Fig. 1) for a graphical representation of cell filtering.

Similar to cell filtering, we defined two criteria for gene filtering: (1) gene expression across all cells of a lineage (excluding cells from knockout and overexpression experiments) exceeds 2000 counts and (2) at least 10 cells from a lineage show gene expression above 10 counts. A gene was filtered if it failed at least one criterion in both lineages. After filtering, count data of 12053 genes across 498 cells remained for further downstream analysis. Remaining count data were normalized by separately applying the sum factor method, as implemented in the R package scater, to cells from the two lineages.

We combined count tables obtained from wild-type cardiac single cells across time points E8.5, E9.5, and E10.5, as well as from Nkx2-5 knockout cardiac cells[19] from E9.5 into a single SCESet object and filtered out cells that were identified as low-quality. After filtering, count data from 11,781 genes across 2358 cells were used to cluster cells using the quickCluster command from the R package scran. Sum factor normalization was applied with deconvolution of size factors within obtained clusters.

Sum factor normalized counts were used to define heterogeneous genes within lineages as well as at individual time points. Specifically, we calculated the coefficient of variation as well as the dropout-rate per gene and investigated their relationship to the mean expression of that gene. We next binned both (ordered) statistics into windows of size 200 and scaled values (z-score transformation) within windows. Genes for which one of the scaled statistics exceeded a 99-percentile within its window where called heterogeneous.

We scaled normalized expression values of heterogeneous genes and used them as input to dimension reduction by self-organizing maps (SOMs) for each lineage. Briefly, SOMs or Kohonen Networks were treated as special cases of neuronal networks, where no target vector containing class labels is necessary for training. Instead, a map is initialized randomly for each cell, consisting of fewer map tiles than input genes, effectively representing meta genes. During training, genes are subsequently placed onto map tiles with the most similar meta gene representation. Importantly, a gene ends upon the same map tile of all cell maps, therefore creating a lower dimensional representation of the cell's transcriptome using meta genes. After 2000 training epochs, cell maps were further projected into two dimensions by t-SNE (perplexity value of 15, 2000 epochs of convergence) and clustered with HDBSCAN using a minimum cluster size of 7 and min_samples 9 (Fig. 1c, d; Supplementary Fig. 2a, b).

Differentially expressed genes between cell clusters were assessed using MAST on sum factor normalized counts (log2 scale). The MAST framework models gene expression in a two-component generalized linear model, one component for the discrete expression rate of each gene across cells and the other component for the continuous expression level, given the gene is expressed. Additionally, we used a gene ranking approach (SC3) to define marker genes specific for each cluster (Supplementary Data Set 1, 2). To define lineage dynamics, we used all protein-coding genes that were marker genes for a cluster (AUROC > 0.8, FDR < 0.01) and differentially expressed in any cluster (lower bound of LFC > 2 or higher bound of LFC < −2, FDR < 0.01) as input to destiny (Fig. 2a, b).

For the critical transition index ($I_C(c)$), we computed the absolute marker gene-to-gene and cell-to-cell correlations for each cluster and calculated the ratio of their means (Supplementary Fig. 6a, b). To reduce influence from differing cell numbers in clusters, we applied a bootstrapping procedure, randomly selecting 30 (20) cells from a given Nkx2-5 lineage cluster or Isl1 lineage cluster repeating the procedure 1000 times. Pairwise cell-to-cell distances were calculated as described by Mohammed and colleagues[28].

To define gene networks that play a role in lineage development, we assumed that genes expression will either increase or decrease with lineage progression. Therefore, we calculated the (global) Spearman's Rank correlation of the expression of each gene to the diffusion pseudotime from destiny. Since a gene might exhibit its expression dynamics only within discrete states (clusters), we also calculated the (local) Spearman's rank correlation of gene expression to pseudotime within clusters. We defined a gene as correlated gene, if it shows a global correlation of at least 0.7 or a local correlation of at least 0.5 (Supplementary Data 3). Lineage-specific correlated genes were used to identify gene networks. Genes within the same sub-network show a high correlation (measured as Pearson's Correlation), but a lower correlation between sub-networks (Supplementary Fig. 7a, b). To identify the dynamics of correlated genes, expression was smoothed along pseudotime by calculating the mean expression in windows of 11 consecutive cells (Supplementary Fig. 7c, d).

To join datasets from two different sequencing platforms, normalized expression values from heterogeneous genes were used as input into the mnnCorrect function from the R package scran. Briefly, mnnCorrect finds cells from different platforms that have mutually similar expression profiles. This is done by identification of pairs of cells that are mutual nearest neighbors, which can be interpreted as belonging to the same cell state. For each MNN pair, the method estimates a pair-specific correction vector. Those vectors are in turn averaged with nearby MNN pair vectors from the same hyperplane using a Gaussian-Kernel to obtain more stable cell-specific correction vectors. The procedure allows correction of cells that are not part of any MNN pair, e.g. data set specific cells that were sampled only on one platform. Corrected expression values were used for clustering and differential expression analysis analogous to steps 6 and 7 (Supplementary Fig. 5).

Cell cycle scores were calculated for each known cell cycle stage (G1/S, S, G2, G2/M, M/G1) using gene sets described by Whitfield et. al[55]. Specifically, a raw score was calculated as the average expression of genes in each set. To refine the

score, we determined genes that correlated (rank correlation > 0.4) well with the raw score and calculated the cell cycle score using those genes. Cell cycle scores were $z$-score transformed (scaled) before plotting. A test of equal proportions was then conducted for cycling cells among Isl1+ and Isl1- cells.

**Single-cell ATAC-seq data analysis**. Annotation of union peaks to nearby transcription start sites was performed using UROPA[56]. Count matrices were stored together with annotation data in a SingleCellExperiment object from the corresponding Bioconductor package. Quality control was done similar to single-cell RNA-seq data (Supplementary Fig. 11b–d): Briefly, a cell-quality map was generated using the per-cell mapping rate, fraction of reads in peaks, fraction of accessible peaks, mitochondrial content, read duplication level, log-scaled total counts, and number of accessible peaks. To define low-quality cells, we evaluated their mitochondrial content, the number of accessible peaks and the log-scaled total-counts using outlier detection with a MADs of 2. Cells failing more than one criterion were considered low-quality and excluded from further analysis.

Similar to cell filtering, we defined two criteria to filter peaks: (1) peak accessibility was given in more than 35 cells and (2) the average count across all cells was below 15, which effectively filters out peaks with exceedingly high coverage. A peak was filtered if it failed at least one criterion. After cell and peak filtering, data remained for 67368 peaks across 695 cells.

To reduce dimensions of the dataset, a binary datum of accessibility was derived by transforming counts greater than 0 to 1 for remaining peaks. Binarized accessibility was used as input for TF-IDF weighting, using term frequency and smoothed inverse document frequency as weighting scheme. Weighted data were reduced to 50 dimensions using SVD. After exclusion of the first SVD dimension, t-SNE was used to project cells into two dimensions (perplexity value of 21, 2000 epochs of convergence, no PCA step). Cells within dense regions were clustered using HDBSCAN with a minimum cluster size of 11 and min_samples of 9.

Peaks were tested for cluster specificity using an empirical Bayes regression-based hypothesis testing procedure implemented in scABC[34]. For each peak, the cluster with lowest resulting $p$-value was chosen as reference cluster. TF-IDF weighted accessibility from cluster-specific peaks with an adjusted p-value lower than 1e-5 was then used as input to destiny to obtain a diffusion pseudo time estimate for cells within each cluster.

To discover transcription factor dynamics and variation in their motif accessibility we conducted analysis using chromVar[35]. Briefly, we downloaded position weight matrices (PWMs) for 579 known TFs from JASPAR and used FIMO with default parameters to find transcription factor motif occurrences in union peaks. Transcription factor motif to peak assignments were used in conjunction with counts from 500 bp size fixed cluster-specific peaks to calculate an accessibility deviation Z-score for each transcription factor motif/cell pair.

**Integrated analysis of single cell RNA and ATAC-seq**. Normalized gene expression values from single-cell RNA-seq data was extracted for TFs with high variability in their motif accessibility across clusters and cells (deviation Z-score > 1.5). Expression values were centered at their mean, ordered by their diffusion pseudo time and smoothed using the mean within a window of size 15 prior to visualization. Similarly, transcription factor motif deviation scores from single-cell ATAC-seq data were ordered by diffusion pseudo time within a cluster and smoothed (window size 13) prior to visualization.

**Bulk ATAC-seq data analysis**. Raw ATAC-seq paired-end reads were trimmed and filtered for quality, and then aligned to the mouse genome GRCm38 (mm10) using STAR[57]. Reads that did not map, mapped non-uniquely, mapped to repetitive regions or to chromosome M, as wells as PCR duplicates were removed.

To remove the non-reproducible replicates, we calculated Spearman correlation using aligned reads, in which the Spearman correlation was above 0.6 resulting in at least two replicates for each developmental stage. For downstream analysis, the read counts were normalized to 1 × depth (reads per genome coverage, RPGC) using the bamCoverage function of deepTools2[58]. Peak calling was performed using callpeak function of MACS2[52] with the following parameters: --nomodel --shift −100 --extsize 200 -q 0.05. Peaks in each sample were merged as union peaks for calculation of peak counts. The normalized number of reads mapped to each peak of the union peaks in each sample was quantified using bigWigAverageOverBed [https://github.com/ENCODE-DCC/kentUtils]. Peak counts of all samples were then merged to obtain a data matrix and normalized with edgeR[59]. Differential accessible peaks were pairwise-compared sequentially across each developmental stage.

The normalized read counts for each developmental stage across replicates were merged, binned around all differential peak summits in 50 bp bins spanning ± 1.5 kb region, clustered by $k$-means algorithm, and visualized by creating heat maps using deepTools2[58].

The proximal and distal peaks are defined by the distance of differential ATAC-seq peaks towards annotated promoters (Gencode annotation): peaks located at least 2.5 kb away from promoters were selected as distal peaks while the others were assigned as proximal peaks.

To compare ATAC-seq peaks with annotated distal *cis*-regulatory elements, the ChIP-seq data of histone modifications H3K4me1, H3K4me3, H3K27ac and H3K27me3 in E10.5 embryonic hearts[39] were downloaded from NCBI Gene

Expression Omnibus (GEO) with the accession ID GSE86753, GSE86752, GSE86723 and GSE86693. The peak density and overlap between ATAC-seq peaks and histone modifications in distal regions (> 2.5 kb to TSS) were calculated using bedGraph files and annotatePeaks function of Homer[40]. The random peaks were generated with the same size distribution as ATAC-seq peaks using shuffleBED function of bedtools.

**Reporting Summary**. Further information on research design is available in the Nature Research Reporting Summary linked to this article.

**Code availability**. The R scripts used for data analysis and simulations are freely available on request or can be downloaded from GitHub [https://github.com/loosolab/cardiac-progenitors].

## Data availability

The authors declare that all data supporting the findings of this study are available within the article and its supplementary information files or from the corresponding author upon reasonable request. All raw and processed data are freely available from the ENA repository and have been deposited under the accession code PRJEB23303. A reporting summary for this article is available as a Supplementary Information file. The source data underlying Figs. 1e, 2c, 2d, 2e, 2f, 5d and 8b and Supplementary Figs 1a–e are provided as a Source Data file.

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

## Acknowledgements

We thank Ann Atzberger for help with cell sorting and Hui Qi for assisting with isolation and dissection of mouse embryos. This work was supported by the Excellence Initiative Cardio-Pulmonary System (ECCPS), the DFG collaborative research centers SFB1213 (TP A02 and B02) and SFB TR81 (TP02), the KFO309 (TP 08), the Foundation Leducq (3CVD01), the German Center for Cardiovascular Research and the European Research Area Network on Cardiovascular Diseases project CLARIFY.

## Author contributions

G.J. and T.B. designed and conceived the project. J.P. and G.J. analyzed the scRNA-seq and bulk RNA-seq data. X.C. performed scATAC-seq. J.P. and X.C. analyzed scATAC-seq data. G.J. analyzed the bulk ATAC-seq data. G.J. isolated, dissected embryos and performed immunofluorescence assays. G.J., S.G., and M.Y. performed scRNA-seq and bulk ATAC-seq. X.Y. provided transgenic mouse lines. S.G., C.K., M.L., Y.Z., and S.T. contributed to data processing, discussions and advice. G.J., J.P., X.C., and T.B. wrote the manuscript.

## Additional information

**Competing interests:** The authors declare no competing interests.

