## [Peer Review File · Nature Communications]

Reviewers' comments:

Reviewer #1 (Remarks to the Author):

In the present manuscript Jia et al. performed single cell RNA-seq experiments for cardiac progenitor cells (CPCs) marked by *Isl1*⁺ or *Nkx2-5*⁺. Using these data, the authors identify subpopulations of two predefined CPC populations. For both CPC populations they describe developmental trajectories. Ablation of *Isl1* in CPC resulted in compromised differentiation and reduced cell cycle activity. Overexpression of *Nkx2-5* in *Isl1*⁺ cells promote the differentiation towards cardiac myocyte-like cells.

They combined these data with bulk RNA-seq and ATAC-seq data for *Isl1*⁺ and *Nkx2-5*⁺ CPCs isolated from different developmental stages. Finally, the authors assessed chromatin accessibility after ablation of *Isl1* or overexpression of *Nkx2-5*.

The presented data is of high technical quality and represents a nice resource for researchers interested in heart development. Main findings are, that *Isl1*⁺ CPC represent a transitory cell population and that *Nkx2-5* primes the cardiac myocyte cell fate. This study clearly supports previous lineage tracing studies.

The manuscript would greatly benefit from application of single cell technology to assess chromatin accessibility.

Major points:

1) The title suggests single cell RNA and single cell ATAC-seq data, which is not the case. The authors should use a more precise title.

2) The description of the used transgenic mouse lines lacks essential details. Especially, the schematic representation of the *Nkx2-5* transgenic line (Fig. 1a) is misleading. According to the cited original publication, this transgenic mouse line was generated using a BAC containing not only the promoter region but also distal regulatory elements.

The used *Isl1* knock-in reporter mice are heterozygous knock-outs for *Isl1*. This genetic alteration may alter the state of *Isl1*-expressing CPC cells? The authors should describe the used mouse strains more thoroughly and highlight their limitations.

3) The strength of single cell analysis is to capture all differentiation states of a progenitor cell population in an unbiased manner. The authors decided to analyze two CPC subpopulations identified by previously described CPC marker genes. However, it remains unclear how these two CPC subsets relate to each other. Are they overlapping or interconnected by several intermediate cell populations? This is an important question, which is also highlighted in the introduction of the manuscript. Therefore, the study would greatly benefit from an unbiased effort to generate CPC trajectories, i.e. by lineage tracing of *Isl1*⁺ cells using a *Isl1*-Cre. Or do subpopulations of *Isl1*⁺ and *Nkx2-5*⁺ CPCs show a common gene expression signature?

Possibly, reanalysis of data from E9.5 hearts published by DeLaughter DM et al. (Dev Cell, 2016) can answer this question, too.

4) Bulk RNA-seq and ATAC-seq data were performed for three developmental time points for *Nkx2-5*⁺ and *Isl1*⁺ CPC. However, the single cell analysis of these time points does not support distinct developmental stages of CPCs at these developmental time points. As stated in the results section *Isl1*⁺ and *Nkx2-5*⁺ subpopulations are not stage-dependent, especially in case of *Isl1*⁺ CPC (results section, page 5 and Fig. 1D). The authors state that "cells collected at the same embryonic stage align to broad pseudotimes". This heterogeneity of CPCs makes the comparison of CPCs isolated from different stages questionable. Therefore, the manuscript would greatly benefit from analysis of chromatin accessibility with single cell resolution. This would allow the clustering of the different CPC subpopulations based on chromatin accessibility in the *Isl1*⁺ and *Nkx2-5*⁺ lineage. Possibly, this strategy would also allow the identification of differential accessible *Isl1* or *Nkx2-5* binding sites in *Isl1*-KO or *Nkx2-5*-OE cells, which likely are present only in selected CPC subpopulations.

5) As expected ATAC peaks are detected in H3K4me3⁺ sites (promoters) as well as H3K4me3⁻/H3K27ac⁺/H3K4me1⁺ sites (enhancers, Fig. 5). The authors should discriminate also in the results section clearly between proximal and distal ATAC peaks.

6) The authors use ChIP-seq data for different histone modifications. However, it remains unclear from which cells these data were obtained. For sure it would be important to use data generated from CPC.

Are these public available data sets or data generated for this study?

7) Among the priming genes defined by the authors are genes of the Hox-family. Is the diminished expression of these genes associated with changes in the chromatin signature?

8) In some figures the symbols/ colors are hard to distinguish (i.e. Fig. 2d, green and blues)

9) Please label the clusters in Fig. 3d.

10) The GREAT analysis of differential peaks upon *Isl1* deletion does not show an unexpected result (Fig. 6C). Does it differ from the distribution of stable peaks in these cells (outside of promoter regions)?

11) The entire results section reads rather technical. The manuscript would benefit from highlighting the affected biological processes. For example, in the first paragraph of the results section the gene ontology analysis of the different cluster should be explained (Suppl. Fig. 4).

12) The legends could be more informative to help the reader to understand the significance of the presented data.

Please rephrase the legend to Fig. 5c. It does not explain the figure (ATAC-expression?).

--

Reviewer #2 (Remarks to the Author):

(Journal) Manuscript Review

MS# NCOMMS-17-31336

Title: Single cell RNA-seq and ATAC-seq indicate critical roles of *Isl1* and *Nkx2-5* for cardiac progenitor cell transition states and lineage settlement

Summary: The authors examined the role of *Nkx2-5* and *Isl1* on cardiac cell differentiation and development by taking advantage of single-cell RNA sequencing (scRNA-seq) and ATAC-seq. Time-course scRNA-seq data identified gradual developmental stages of *Nkx2-5*⁺ cells and bifurcating differentiation of *Isl1*⁺ progenitor cells into cardiomyocyte or endothelial cells. Even though similar studies have been performed for decades using lineage tracing and clonal analysis, the superior point of this study comparing to previous works is that this study exclusively focused on *Nkx2-5* expressing cells instead of derivatives of those, enabling to show a clear role of those cells in cardiac development. In addition, using ATAC-seq technology, the authors showed changes in chromatin accessibility by *Isl1* and *Nkx2-5*.

Comments on Conceptual Novelty and impact on the field:

scRNA-seq with embryonic cardiac cells were performed in previously published studies. Christine Seidman group reported developmental changes in single cell transcriptome from embryo to postnatal (Dev Cell. 2016 39: 480–490) and back-to-back paper by Sean Wu developed Anatomical Transcription-based Legend from Analysis of Single-cell RNA-Sequencing (ATLAS-seq) by dissecting embryonic heart cells followed by scRNA-seq, identifying chamber-specific genes (Dev Cell. 2016 39: 491–507). To my knowledge, this manuscript is the first paper that systematically performed transcriptomic profiling of *Nkx2-5*⁺ or *Isl1*⁺ cells from developmental heart at single-cell level (from E7.5 to E12.5), demonstrating clear developmental trajectories of those cardiac progenitor cells. And their ATAC-seq data suggest a mechanistic explanation how *Nkx2-5* and *Isl1* work in cardiogenesis differently.

Minor issues

1. Nkx2-5+ cluster 3 shows low expression in cTnT and a-SMA (Fig1e and f), but in the result section, it was mentioned that it is characterized by (high) expression level of cTnT and a-SMA. Is that a mistake by confusing between cluster 1 and 3 in the text? Please clarify whether it is just typos or a big mistake through the scRNA-seq analysis part.
2. Three different blue colors used for left panel of Fig1d and thereafter don't have enough contrast between them, especially colors for E8.5 and E9.5 are not distinguishable at all on a paper. Change the colors to secure a clear contrast. The green colors in right panel of Fig1d have the same problem.
3. Fig1c and Fig2a represent that cluster 3 -> cluster 2 -> cluster 1 is the developmental order, but Fig1d seems to represent that cluster 3-> cluster1 -> cluster 2. How do you explain the mismatched pattern?
4. Supple figure 4b doesn't represent differences between three clusters; all the GO terms are related with muscle development and contraction. However, authors argued that only one of those three Nkx2-5+ cell cluster represents myogenic fate. Supple figure 4b is not convincing.

--

Reviewer #3 (Remarks to the Author):

The paper by Jia and Braun and colleagues is an elegant example of how single cell transcriptomics and chromatin accessibility assays can be applied to understand key developmental transitions. Nkx2-5 and Isl1 have been well studied genetically, and positive populations have been analysed by lineage tracing, revealing many new insights into heart development, although the fine molecular regulation of gene networks by these factors is unknown. The study has used a combination of time resolved scRNAseq on WT and knockout or over-expression tissue, and ATACseq on bulk populations, to seek new network insights. These include the general pseudo-time features of captured cell population and chromatin dynamics associated with the transition from CPCs to CMs and ECs - this provides a valuable resource for future studies; specific features of the CPC-CM/EC transition that concur with transition state; specific network roles of Nkx2-5 versus Isl1; direct versus indirect action of these TFs; and novel pathways and gene that govern this phase of development. Many findings make good sense relative to what we already know about this process, eg role of Mef2c in CM differentiation etc, and the general requirement to shut down progenitor networks during the CPC differentiation. The role of other genes/pathways highlighted will have to be confirmed with experimental studies. The highest-level findings relate to the different regions of the genome that are opened and closed during differentiation from CPC, how these differ and overlap with respect to Isl1 and Nkx2-5; the implications for gene expression and network control, and how chromatin organisation precedes specific network change. Overall this is a valuable contribution to cardiac network biology.

Comments:

1. KO of Isl1 and over-expression of Nkx2-5 was performed leading to valuable insights. However, analysis of Nkx2-5 OE hearts was performed at E12.5 – some caution is needed then in comparing to data from E9.5. The authors should be very explicit about the caveats. On page 10 end of second paragraph, the authors conclude from analysis of GFP+ cells at E12.5 that Nkx2-5 is required and sufficient for resolution of the multipotent differentiation capacity of CPC. This is based on the observation that OE cells align on the CM Isl1 pseudotime trajectory and not along the EC cell branch. However, Isl1 may turn off after differentiation to ECs out to E12.5: how does this affect the conclusion?
2. I did not feel there was sufficient clarity around how data should be interpreted knowing that there will be a population of Nkx2-5 cells also expressing Isl1 and visa versa, and that this ratio may change over time. How might this affect the results and conclusions?
3. How is ATACseq limited by the fact that it was applied to bulk populations which contain mixtures of CPC and more differentiated cells that change over time. How might this affect the results and conclusions?

4. It is difficult to keep track of the different populations identified by tSNE analysis as the narrative unfolds. It would be help full to further label figures to indicate the key populations under discussion and their qualifiers – eg. EC, CM, CPCearly/late etc.
5. Page 5. I am confused by the assignment of Nkx2-5 cluster 3 as a CPC being from E7.5. This does not seem to fit with the pseudotrajectory arrow in Fig. 2A and the finding that this cluster express higher levels of CM markers than the other later clusters in particular cluster 1 (Fig. 1f – cTnC, Sma).
6. It should be acknowledged that Nkx2-5 is also expressed in a subset of endocardial cells. These are not captured, likely because Nkx2-5 and GFP are not robustly expressed there.
7. Page 6 para 3. It says here that Nkx2-5 clusters 3 and 1 correspond to E8.5 and E9.5 respectively. See above. Is this a mistake?
8. Does it seem surprising that clusters seem very discrete and that cells bridging clusters are not generally evident. Can discrete clusters be claimed given so few cells profiled? Are there insights from the replication experiment with iCell8 platform.
9. Page 7: what is meant by the conclusion that CPS are not synchronized at different embryonic stages but follow individual developmental traits?
10. Why would Ic(c) values be higher or equal in transition states than in differentiated states. Further rationale and assumptions need to be incorporated into this section.
11. Page 13 sentence 2. Add “compare to WT”? But how can this be determined given that the inactivation embryos were analysed only at one time point – E9.5?
12. Page 13: “...92% of the peaks...presumably representing enhancer regions affecting progenitor cell decisions.” Unclear.
13. Page 13: “We concluded that Isl1 acts together with Mef2 factors but prevents binding of Forkhead factors...” Not sure this fits with what was said previously.
14. Page 14: “..forced expression of Nkx2-5 resulted in a dramatic increase of accessible chromatin sites at E9.5 compared to E12.5..” Don't the authors mean E12.5 relative to E9.5? See above.

--

Reviewer #4 (Remarks to the Author):

This is a nicely written and comprehensive study about cardiac progenitor cell fate decisions. Overall, I have a positive view about this manuscript, but it needs some improvements to be acceptable for publishing. In particular, there are a number of conclusions that are not supported by the experiments/analysis performed. Here are my specific comments:

1) It remains unclear how many embryos per stage were used in experiments shown in Figure 1 and 2. If the trajectory and all the subsequent analyses were generated based on the data from one embryo from each stage, then authors should include data from more embryos using the same experimental approach so that data can be combined and reanalysed. This would give confidence in the robustness of the conclusions.

Since the data generated from three different developmental stages were combined how did the authors deal with batch effects? Especially in the case of Nkx-2 clustering where it appears that two out of three clusters contain cells from distinct developmental stages.

2) Authors stated the following:

“Projection of Isl1-KO single cells on the trajectory of the developing SHF revealed that Isl1-KO cells are stalled/trapped in the previously identified stable attractor state (Fig. 3b).”

This is a 2D projection. The Isl1-KO cells might be perpendicular (or far away) to the wt cells in the 3D space. Authors should show data in 3D so that it is clear that they are really occupying the

same attractor state.

3) Authors stated the following:

“Our pseudotime-based analysis of developmental trajectories revealed one continuous trajectory of Nkx2-5+ CPC suggesting that Nkx2-5+ cells are exclusively committed to become cardiomyocytes. In contrast, the trajectory of Isl1+ CPC bifurcated into two distinct orientations (endothelial cells and cardiomyocytes), suggesting the existence of a transition state, which separates multipotency of Isl1+ CPC from acquisition of distinct cellular identities (Fig. 2b).”

It is not correct to make such a statement just based on the way trajectory looks like. These two trajectories (Nkx2-5+ and Isl1+) look very similar (in terms of overall shape) and only the subsequent analysis revealed the differences between them. Therefore, this part should be rewritten and conclusions should be made later on in the manuscript when all the evidence for such a claim has been laid out.

An important point here is that the lack of cells with the endothelial identity in Nkx2-5+ trajectory is not the proof of their unilineage potential i.e. committed state. As authors noted themselves lack of expression of Nkx2-5 in the endothelial cells might have prevented their perspective sorting. In fact, there is a compelling evidence that some of Nkx2-5+ cells are multipotent. First, the lineage tracing studies have shown the multipotency of Nkx2-5+ cells and second the endogenous expression of Nkx2-5 in cluster 3 is close to zero.

Thus, the authors should not make such claims unless they are willing to do functional experiments to prove that indeed Nkx2-5+ are already committed. Instead, I would suggest that they point out limitations of their system (e.g. down regulation of Nkx2 in endothelial cells) and move on to show the functional relevance of this gene in the cardiomyocyte differentiation. This has been nicely demonstrated in their Nkx2-5 expression experiment.

5) Authors stated the following:

Since our scRNA-seq analysis at E9.5 indicated arrest of Isl1 mutant CPC development, essentially converting the transcriptional profile of E9.5 into an E8.5 state (Fig. 3), we compared the landscape of chromatin accessibility of both populations. Surprisingly, we found that E8.5 Isl1+ cells and E9.5 Isl1 mutant cells show significantly different open chromatin signatures (432 opening and 728 closing), although they share similar positions at the developmental pseudotime trajectory (Fig. 6a), which indicates that Isl1-dependent changes in chromatin accessibility occur ahead of transcriptional divergence.

I do not see the evidence that *isl1* mutant CPC have been transcriptionally converted from E9.5 to E8.5. First, although *isl1*KO cells cluster together their relationship to *isl1*+ cells cannot be assessed based on the 2D projection that authors show. It would be more appropriate to combine data from E7.5, E8.5 and E9.5 *Isl1*+ with E9.5 *Isl1* mutant and perform clustering (similar to the one in Figure 1). The expectation is that E8.5 *Isl1*+ cells and E9.5 *Isl1* mutant cells would cluster together (if they are transcriptionally similar they should be intermixed). The actual transcriptional differences between E8.5 *Isl1*+ cells and E9.5 *Isl1* mutant cells could be assessed by performing differential expression analysis. Second, *isl1*+ cells from E9.5 and E8.5 are intermixed in pseudotime and sit close to each other at the beginning of the trajectory. On the page 5 authors stated: “Stage-dependent clustering was less evident for the five *Isl1*+ subpopulations....”. Therefore, I disagree with the logic to compare open chromatin signatures of E8.5 *Isl1*+ cells and E9.5 *Isl1* mutant cells and the statement that they share similar positions at the developmental pseudotime trajectory.

Minor point – authors should change the colours of cells at different stages of development. At the moment it is hard to see distribution of cells in pseudotime because different shading of the same

colour is used.

Response to reviewers

Reviewer #1:

In the present manuscript Jia et al. performed single cell RNA-seq experiments for cardiac progenitor cells (CPCs) marked by Isl1+ or Nkx2-5+. Using these data, the authors identify subpopulations of two predefined CPC populations. For both CPC populations they describe developmental trajectories. Ablation of Isl1 in CPC resulted in compromised differentiation and reduced cell cycle activity. Overexpression of Nkx2-5 in Isl1+ cells promote the differentiation towards cardiac myocyte-like cells.

They combined these data with bulk RNA-seq and ATAC-seq data for Isl1+ and Nkx2-5+ CPCs isolated from different developmental stages. Finally, the authors assessed chromatin accessibility after ablation of Isl1 or overexpression of Nkx2-5.

The presented data is of high technical quality and represents a nice resource for researchers interested in heart development. Main findings are, that Isl1+ CPC represent a transitory cell population and that Nkx2-5 primes the cardiac myocyte cell fate. This study clearly supports previous lineage tracing studies.

The manuscript would greatly benefit from application of single cell technology to assess chromatin accessibility.

Response: We thank the reviewer for the comments and helpful suggestions. To comply with the reviewer's request we performed chromatin accessibility assays in single Isl1⁺ CPCs (single cell ATAC-seq or scATAC-seq). The data set provides additional important insights into the heterogeneity of CPCs. Importantly, the single cell chromatin accessibility profiles resemble the transcriptional heterogeneity of Isl1⁺ CPCs. We think that we have greatly strengthened our conclusions by confirming the existence of different CPC sub-population via scATAC-seq. The scATAC-seq essentially recapitulated the cardiomyocyte and endothelial lineage bias demonstrated by our scRNA-seq and enabled us to explore the TF binding dynamics in different developmental trajectories.

Major points:

1) The title suggests single cell RNA and single cell ATAC-seq data, which is not the case. The authors should use a more precise title.

Response: We agree that the original title without performing single cell ATAC-seq analysis might have been misleading. However, since we now include single cell ATAC-seq data, we reason that the title is appropriate. We hope the reviewer concurs.

2) The description of the used transgenic mouse lines lacks essential details. Especially, the schematic representation of the Nkx2-5 transgenic line (Fig. 1a) is misleading. According to the cited original publication, this transgenic mouse line was generated using a BAC containing not only the promoter region but also distal regulatory elements.

The used Isl1 knock-in reporter mice are heterozygous knock-outs for Isl1. This genetic alteration may alter the state of Isl1-expressing CPC cells? The authors should describe the used mouse strains more thoroughly and highlight their limitations.

Response: We thank the reviewer for these comments. We have now accurately depicted the generation of the Nkx2-5 transgenic line in Fig. 1a as requested. The Isl1 knock-in reporter indeed shows minor effects on Isl1 expression levels. However, as reported in our previous paper [1], this does not cause any apparent defects during development and in adult stages. To address this issue, we now write:

“Insertion of the GFP-reporter gene into one allele of the Isl1 gene had minor but measurable effects on Isl1 expression levels. However, we did not observe any apparent defects during cardiac development and in adult stages [9]. The Nkx2-5-emGFP transgenic mouse line was generated using a BAC containing both the promoter region and distal regulatory elements, which enables faithful recapitulation of Nkx2-5 expression [10]. [...]” (Results, Paragraph 1).

3) The strength of single cell analysis is to capture all differentiation states of a progenitor cell population in an unbiased manner. The authors decided to analyze two CPC

subpopulations identified by previously described CPC marker genes. However, it remains unclear how these two CPC subsets relate to each other. Are they overlapping or interconnected by several intermediate cell populations? This is an important question, which is also highlighted in the introduction of the manuscript. Therefore, the study would greatly benefit from an unbiased effort to generate CPC trajectories, i.e. by lineage tracing of *Isl1*⁺ cells using a *Isl1*-Cre. Or do subpopulations of *Isl1*⁺ and *Nkx2-5*⁺ CPCs show a common gene expression signature?

Possibly, reanalysis of data from E9.5 hearts published by DeLaughter DM et al. (Dev Cell, 2016) can answer this question, too.

Response: We thank the reviewer for drawing our attention to this issue. We agree that it is important to study the relation between *Isl1* and *Nkx2-5* expressing cells as well as potential overlaps between both populations and/or intermediate cell populations. Although lineage tracing is a powerful tool to determine the origin of distinct cell population, we are not sure whether its use makes sense in the current setting, which analyses the heterogeneity of cardiac progenitor cells using scRNA-seq and scATAC-seq. Heterogeneity might also arise when different populations share a common origin. We did not focus on the origin of cells but specifically on actual transcriptional patterns (and chromatin accessibility) in individual cells to generate CPC trajectories. Only the transcriptional analysis allows identification of genes, which drive lineage segregation.

Nevertheless, cells co-expressing *Isl1*⁺ and *Nkx2-5*⁺ CPCs might cause confounding effects, which might affect subsequent analysis. Therefore, we performed co-immunofluorescence staining for *Isl1* and *Nkx2-5* in E8.5 and E9.5 embryos. The majority of either *Isl1*⁺ or *Nkx2-5*⁺ CPCs located in different regions at these stages of development (namely the first heart field and second heart field) (Supplementary Figure 8). However, we observed co-expression within the coelomic walls and the proximal head mesenchyme, which has also been reported in previous studies. To further refine the analysis, we generated compound reporter mouse lines, in which *Isl1*-expression is indicated by nuclear localization of GFP fused to H2B (*Isl1*^{+/nGFP}), whereas *Nkx2-5* expression is indicated by the presence of emGFP in the cytoplasm (Fig. 3a, b). We detected a small population of *Isl1* and *Nkx2-5* co-expressing cells, although the majority expressed either *Isl1* or *Nkx2-5* (Fig. 3b).

We also followed the reviewer's advice and determined whether *Isl1*⁺/*Nkx2-5*⁺ CPCs show a common gene expression signature that differs from cells that are either *Isl1*⁺ or *Nkx2-5*⁺ along with pseudotime analysis of *Isl1*⁺/*Nkx2-5*⁺ CPCs (Fig. 3d, e; Fig. 4). Analysis of scRNA-seq data revealed that there are only few *Isl1*⁺/*Nkx2-5*⁺ cells, which is consistent with the immunofluorescence staining of *Isl1*^{+/nGFP}/*Nkx2-5*-emGFP double positive mice. In general, we detected fewer cells with *Isl1* and *Nkx2-5* transcripts in the *Isl1* lineage (defined by *Isl1*^{+/nGFP}) compared to the *Nkx2-5* lineage. In addition, the expression level of *Nkx2-5* was relatively low. In contrast, the *Nkx2-5* lineage (defined by *Nkx2-5*-emGFP) contained more double positive cells and the expression level of *Isl1* was higher. When we took the *Isl1* diffusion map and positioned *Nkx2-5* expressing cells onto the map, we were able to classify several categories: (1) *Isl1*⁺/*Nkx2-5*⁻, (2) *Isl1*⁻/*Nkx2-5*⁺, and (3) cells with varying levels of both *Isl1* and *Nkx2-5*. Co-expressing cells were color-coded, more green when *Isl1* was expressed at higher levels and more red when *Nkx2-5* was expressed at higher levels, (4) double negative cells. In Fig. 3d, we only considered cells from the *Isl1*-lineage and crossed out cells from the *Nkx2-5* lineage. In Fig. 3e, cells from *Nkx2-5* lineage were colored and cells from *Isl1*-lineage were crossed out. Consistently, *Nkx2-5* is only expressed in cells that fit on the trajectory towards cardiomyocytes, irrespectively of whether the cells co-express *Isl1*. We can even detect a switch-like expression at the end of the trajectory. In contrast, cells are *Isl1* positive but *Nkx2-5* negative (all cells are labeled green) during the transition state. Early cells in the *Nkx2-5* trajectory also express *Isl1*, but to a lower extent, suggesting that early *Nkx2-5* expression primes CPCs towards a cardiac fate. Furthermore, *Nkx2-5* primed cells seem to take a shortcut to cardiomyocytes, since they do not overlap with the *Isl1* transition state.

The detection of double negative cells (i.e. cells, in which neither *Isl1* nor *Nkx2-5* transcripts were detected), mostly at early (E7.5) stages, seems counterintuitive, since all CPCs were isolated based on reporter gene activity driven by the *Isl1* or *Nkx2-5* promoters. However, such a phenomenon is not unusual and most likely reflects transcriptional fluctuations that are not accurately represented by the GFP proteins, which are more stable than the mRNAs. Previous studies demonstrated that mRNAs of genes controlling cell-type specific functions in a limited time window usually have short life spans (<2-3h) and show fluctuations of transcription [2]. Double negative cells, probably reflecting such fluctuations were primarily detected in early CPCs progenitors when transcription of *Isl1* and *Nkx2-5* becomes established. Of note, removal of double negative cells did not affect the outcome of the clustering. We therefore did not discard such cells, since we believe they reflect the real transcriptional dynamics/fluctuation of *Isl1* and *Nkx2-5* and contribute to the heterogeneity.

Direct comparison of *Isl1*⁺, *Nkx2-5*⁺CPCs or *Isl1*⁺/*Nkx2-5*⁺ cells did not yield a distinct cluster that would not fit into *Isl1*⁺ or *Nkx2-5*⁺ clusters further suggesting that double positive cells do not follow a distinct fate but represent a temporary state along the trajectories (Fig. 3c-e). To further highlight the differences between *Isl1* and *Nkx2-5* expressing cells, we performed in situ hybridization of a few candidate genes, which were found to be differentially expressed in our bioinformatics analysis (Fig. 4).

4) Bulk RNA-seq and ATAC-seq data were performed for three developmental time points for *Nkx2-5*⁺ and *Isl1*⁺ CPC. However, the single cell analysis of these time points does not support distinct developmental stages of CPCs at these developmental time points. As stated in the results section *Isl1*⁺ and *Nkx2-5*⁺ subpopulations are not stage-dependent, especially in case of *Isl1*⁺ CPC (results section, page 5 and Fig. 1D). The authors state that “cells collected at the same embryonic stage align to broad pseudotimes”. This heterogeneity of CPCs makes the comparison of CPCs isolated from different stages questionable. Therefore, the manuscript would greatly benefit from analysis of chromatin accessibility with single cell resolution. This would allow the clustering of the different CPC subpopulations based on chromatin accessibility in the *Isl1*⁺ and *Nkx2-5*⁺ lineage. Possibly, this strategy would also allow the identification of differential accessible *Isl1* or *Nkx2-5* binding sites in *Isl1*-KO or *Nkx2-5*-OE cells, which likely are present only in selected CPC subpopulations.

Response: We thank the reviewer for this suggestion. We have performed single ATAC-seq on *Isl1*⁺ CPCs, which essentially corroborate the results obtained by scRNA-seq. In addition, we separated the *Isl1*⁺ sub-populations at E9.5 and E8.5 by FACS using a CD31 antibody and performed bulk ATAC-seq. This experiment allowed us to determine more precisely the chromatin accessibility profiles in the *Isl1*⁺CD31⁻ (cardiomyocyte branch) versus the *Isl1*⁺CD31⁺ (endothelial branch) subpopulations. Next, we compared the new ATAC-seq data from *Isl1*⁺CD31⁻ and *Isl1*⁺CD31⁺ cells to those from *Isl1* KO and *Nkx2-5*-OE cells. Interestingly, inactivation of the *Isl1* gene did not cause substantial changes compared to *Isl1*⁺/CD31⁻ CPCs at E8.5 or E9.5 (Fig. 9a, b). Loss of *Isl1* led to more robust closing than opening of chromatin regions compared to *Isl1*⁺/CD31⁺ CPCs at E8.5 and E9.5 suggesting that *Isl1* is required to leave the attractor state characterized by an open chromatin organization (Fig. 9a & b). Taken together, these results indicate that *Isl1* mutant CPCs exhibit an epigenomic profile that resembles developing cardiomyocytes and differs from endothelial cells.

5) As expected ATAC peaks are detected in H3K4me3⁺ sites (promoters) as well as H3K4me3⁻/H3K27ac⁺/H3K4me1⁺ sites (enhancers, Fig. 5). The authors should discriminate also in the results section clearly between proximal and distal ATAC peaks.

Response: We have improved the description of criteria used to discriminate proximal and distal regulatory regions in both the main text and the Methods sections (see below).

“The proximal and distal peaks were defined by the distance of differential ATAC-seq peaks towards annotated promoters (Gencode annotation): peaks at least 2.5 kb away from transcriptional start sites were defined as distal peaks, peaks closer to transcriptional start sites were defined as proximal peaks.”

We would also like to mention in this context that the additional comparison of ATAC-seq data from *Isl1*⁺CD31⁺ and *Isl1*⁺CD31⁻ cells to *Isl1* gene knockout cells revealed little changes between *Isl1* knockout cells and *Isl1*⁺CD31⁻ cells but substantial differences between *Isl1* knockout cells and *Isl1*⁺CD31⁺ cells, which were mostly (87.5%) co-localized with H3K4me3⁻/H3K27ac⁺/H3K4me1⁺ sites (i.e. presumptive enhancers). These new data are shown in (Supplementary Figure 13e).

6) The authors use ChIP-seq data for different histone modifications. However, it remains unclear from which cells these data were obtained. For sure it would be important to use data generated from CPC. Are this public available data sets or data generated for this study?

Response: We are sorry for this omission. The data were derived from the ENCODE project, in which ChIP-seq data for individual histone modifications were determined using mouse embryonic hearts at E10.5 [3]. We have added this missing information. To the best of our knowledge, histone modifications have not been assessed in cardiac progenitor cells so far due to technique difficulties

(limited number of cells). Nevertheless, we think it is reasonable to assume that the majority of tissue specific enhancers are relatively stable. To cope with these limitations, we have added a sentence pointing out the limitations.

7) Among the priming genes defined by the authors are genes of the Hox-family. Is the diminished expression of these genes associated with changes in the chromatin signature?

Response: Thanks to the single cell ATAC-seq analysis, we can clearly say “Yes”. The scATAC-seq data show enrichment of Hox motifs in Isl1⁺ CPCs associated with settlement of the cardiac lineage (Fig. 8b, e & f).

8) In some figures the symbols/ colors are hard to distinguish (i.e. Fig. 2d, green and blues)

Response: We thank the reviewer for pointing out these issues. We have changed the colors in the figures accordingly.

9) Please label the clusters in Fig. 3d.

Response: We have added names to clusters using qualifier term “CPC intermediate and late” together with numbers. The previous Fig. 3d now is Fig. 5d.

10) The GREAT analysis of differential peaks upon Isl1 deletion does not show an unexpected result (Fig. 6C). Does it differ from the distribution of stable peaks in these cells (outside of promoter regions)?

Response: We performed additional experiments in which we separated E9.5 and E8.5 Isl1⁺CD31⁻ (cardiomyocyte trajectory) from Isl1⁺CD31⁺ (endothelial trajectory) cells and performed bulk ATAC-seq followed by comparison to ATAC seq data obtained from Isl1 KO cells. We observed robust changes when comparing Isl1 KO to Isl1⁺CD31⁺ cells but little changes when comparing to Isl1⁺CD31⁻ cells. In this setting, GREAT analysis consistently detected the majority of differential peaks in distal regions, which were then interrogated in respect to TF motif enrichment. The initial GREAT analysis of stable peaks between Isl1-KO and wild type cells was done with cells that were not separated into CD31⁻ and CD31⁺ fractions. Since we did not detect significant changes of the peaks that were localized to distal regions, we did not to include this information.

11) The entire results section reads rather technical. The manuscript would benefit from highlighting the affected biological processes. For example, in the first paragraph of the results section the gene ontology analysis of the different cluster should be explained (Suppl. Fig. 4).

Response: We are aware that the results section reads rather technical. We tried our best to highlight the affected biological processes and added the information about GO terms. Yet, the analysis is primarily driven by computational approaches, which necessarily affects the narrative.

12) The legends could be more informative to help the reader to understand the significance of the presented data.

Please rephrase the legend to Fig. 5c. It does not explain the figure (ATAC-expression?).

Response: We agree that it helps the readers to explain the results in the legends rather than to simply describe the figure. Most journals explicitly ask to describe but not interpret the figures in the legends. We tried to extend the descriptions in the legends a bit without repeating of what is already said in the main text. The old Fig. 5c is now Supplementary Fig.13d and the legend was modified by incorporating an explanation for the metagene plot. Essentially, the normalized enrichment of ATAC-seq signals was aggregated based on the positions of ATAC-seq signals relative to all genes with the purpose to generate a visual representation of the general pattern.

Reviewer #2 (Remarks to the Author):

Title: Single cell RNA-seq and ATAC-seq indicate critical roles of Isl1 and Nkx2-5 for cardiac progenitor cell transition states and lineage settlement

Summary: The authors examined the role of Nkx2-5 and Isl1 on cardiac cell differentiation and development by taking advantage of single-cell RNA sequencing (scRNA-seq) and ATAC-seq. Time-course scRNA-seq data identified gradual developmental stages of Nkx2-5+ cells and bifurcating differentiation of Isl1+ progenitor cells into cardiomyocyte or endothelial cells. Even though the similar studies have been performed for decades using lineage tracing and clonal analysis, the superior point of this study comparing to previous works is that this study exclusively focused on Nkx2-5 expressing cells instead of derivatives of those, enabling to show a clear role of those cells in cardiac development. In addition, using ATAC-seq technology, the authors showed changes in chromatin accessibility by Isl1 and Nkx2-5.

Comments on Conceptual Novelty and impact on the field:

scRNA-seq with embryonic cardiac cells were performed in previously published studies. Christine Seidman group reported developmental changes in single cell transcriptome from embryo to postnatal (Dev Cell. 2016 39: 480–490) and back-to-back paper by Sean Wu developed Anatomical Transcription-based Legend from Analysis of Single-cell RNA-Sequencing (ATLAS-seq) by dissecting embryonic heart cells followed by scRNA-seq, identifying chamber-specific genes (Dev Cell. 2016 39: 491–507). To my knowledge, this manuscript is the first paper that systematically performed transcriptomic profiling of Nkx2-5+ or Isl1+ cells from developmental heart at single-cell level (from E7.5 to E12.5), demonstrating clear developmental trajectories of those cardiac progenitor cells. And their ATAC-seq data suggest a mechanistic explanation how Nkx2-5 and Isl1 work in cardiogenesis differently.

Response: We thank the reviewer for the supportive comments and helpful suggestions.

Minor issues:

1. Nkx2-5+ cluster 3 shows low expression in cTnT and a-SMA (Fig1e and f), but in the result section, it was mentioned that it is characterized by (high) expression level of cTnT and a-SMA. Is that a mistake by confusing between cluster 1 and 3 in the text? Please clarify whether it is just typos or a big mistake through the scRNA-seq analysis part.

Response: We thank the reviewer for pointing out the inconsistency. The reviewer is right. This is a typo, which we have corrected.

2. Three different blue colors used for left panel of Fig1d and thereafter don't have enough contrast between them, especially colors for E8.5 and E9.5 are not distinguishable at all on a paper. Change the colors to secure a clear contrast. The green colors in right panel of Fig1d have the same problem.

Response: We have changed the colors, which was also suggested by Reviewer #1 (Point 8).

3. Fig1c and Fig2a represent that cluster 3 -> cluster 2 -> cluster 1 is the developmental order, but Fig1d seems to represent that cluster 3-> cluster1 -> cluster 2. How do you explain the mismatched pattern?

Response: Our analysis clearly demonstrates that development of Isl1+ or Nkx2-5+ cells is not synchronized at any given time point. With other words: there are already some cells at E8.5 that are more mature than others. The same is true for E9.5 cells: some cells are less mature than others, resembling more E8.5 cells. This phenomenon can give rise to the seemingly paradox situation pointed out by the reviewer. In our view, the stages at which the cells were isolated do not necessarily reflect the “true” developmental stages. We have added a sentence to better explain the situation.

4. Supple figure 4b doesn't represent differences between three clusters; all the GO terms are related with muscle development and contraction. However, authors argued that only one of those three Nkx2-5+ cell cluster represents myogenic fate. Supple figure 4b is not convincing.

Response: We are sorry for the misunderstanding. Suppl. Figure 4b was meant to demonstrate that all three clusters of Nkx2-5⁺ cells already express myogenic genes, leading to the enrichment of GO terms for muscle development and contraction. However, these clusters also contain other differentially expressed genes not shown in the figure, which allowed us to draw conclusion about the developmental trajectory. We have added the following sentence to explain the purpose of the figure in the manuscript “In addition to numerous differentially expressed genes, all three clusters of Nkx2-5⁺ CPCs show an enrichment of GO terms related to muscle development and contraction (Supplementary Fig. 4b) suggesting that Nkx2-5 is associated with myogenic differentiation while Isl1 is linked to the maintenance of progenitor cell multipotency. [...]” (Page 6).

Reviewer #3 (Remarks to the Author):

The paper by Jia and Braun and colleagues is an elegant example of how single cell transcriptomics and chromatin accessibility assays can be applied to understand key developmental transitions. Nkx2-5 and Isl1 have been well studied genetically, and positive populations have been analysed by lineage tracing, revealing many new insights into heart development, although the fine molecular regulation of gene networks by these factors is unknown. The study has used a combination of time resolved scRNAseq on WT and knockout or over-expression tissue, and ATACseq on bulk populations, to seek new network insights. These include the general pseudo-time features of captures cell population and chromatin dynamics associated with the transition from CPCs to CMs and ECs - this provides a valuable resource for future studies; specific features of the CPC-CM/EC transition that concur with transition state; specific network roles of Nkx2-5 versus Isl1; direct versus indirect action of these

TFs; and novel pathways and gene that govern this phase of development. Many findings make good sense relative to what we already know about this process, eg role of Mef2c in CM differentiation etc, and the general requirement to shut down progenitor networks during the CPC differentiation. The role of other genes/pathways highlighted will have to be confirmed with experimental studies. The highest-level findings relate to the different regions of the genome that are opened and closed during differentiation from CPC, how these differ and overlap with respect to Isl1 and Nkx2-5; the implications for gene expression and network control, and how chromatin organisation precedes specific network change. Overall this is a valuable contribution to cardiac network biology.

We thank the reviewer for the supportive comments and helpful suggestions.

Comments:

1. KO of Isl1 and over-expression of Nkx2-5 was performed leading to valuable insights. However, analysis of Nkx2-5 OE hearts was performed at E12.5 – some caution is needed then in comparing to data from E9.5. The authors should be very explicit about the caveats. On page 10 end of second paragraph, the authors conclude from analysis of GFP+ cells at E12.5 that Nkx2-5 is required and sufficient for resolution of the multipotent differentiation capacity of CPC. This is based on the observation that OE cells align on the CM Isl1 pseudotime trajectory and not along the EC cell branch. However, Isl1 may turn off after differentiation to ECs out to E12.5: how does this affect the conclusion?

Response: We do not think that it matters whether Isl1 is turned off after differentiation to ECs at E12.5, since we used an Isl1-cre line to express Nkx2-5 AND GFP from the Rosa26 locus in Isl1⁺ cells. Once the recombination has occurred, it does not matter anymore when Isl1 is inactivated, since cells that have once expressed Isl1 will continue to express both Nkx2-5 and GFP. The reviewer is right that we might have run into trouble when using a GFP reporter for cell isolation that is directly driven from the Isl1-promoter, which was not the case in this specific experiment. Therefore, we will still capture such cells at E12.5, although Isl1 expression might have been turned off.

2. I did not feel there was sufficient clarity around how data should be interpreted knowing that there will be a population of Nkx2-5 cells also expressing Isl1 and visa versa, and that this ratio may change over time. How might this affect the results and conclusions?

Response: We thank the reviewer for the comment about Nkx2-5 and Isl1 co-expressing cells, which was also raised by Reviewer #1. We have extensively addressed this point in our response to Reviewer #1 Point 3, which we would like to repeat here.

We thank the reviewer for drawing our attention to this issue. We agree that it is important to study the relation between Isl1 and Nkx2-5 expressing cells as well as potential overlaps between both populations and/or intermediate cell populations. Although lineage tracing is a powerful tool to determine the origin of distinct cell population, we are not sure whether its use makes sense in the current setting, which analyses the heterogeneity of cardiac progenitor cells scRNA-seq and scATAC-seq. Heterogeneity might also arise when different populations share a common origin. We did not focus on the origin of cells but specifically on actual transcriptional patterns (and chromatin accessibility) in individual cells to generate CPC trajectories. Only the transcriptional analysis allows identification of genes, which drive lineage segregation.

Nevertheless, cells co-expressing Isl1⁺ and Nkx2-5⁺ CPCs might cause confounding effects, which might affect subsequent analysis. Therefore, we performed co-immunofluorescence staining for Isl1

and *Nkx2-5* in E8.5 and E9.5 embryos. The majority of either *Isl1*⁺ or *Nkx2-5*⁺ CPCs located in different regions at these stages of development (namely the first heart field and second heart field) (Supplementary Figure 8). However, we observed co-expression within the coelomic walls and the proximal head mesenchyme, which has also been reported in previous studies. To further refine the analysis, we generated compound reporter mouse lines, in which *Isl1*-expression is indicated by nuclear localization of GFP fused to H2B (*Isl1*^{+/nGFP}), whereas *Nkx2-5* expression is indicated by the presence of emGFP in the cytoplasm (Fig. 3a, b). We detected a small population of *Isl1* and *Nkx2-5* co-expressing cells, although the majority expressed either *Isl1* or *Nkx2-5* (Fig. 3b).

We also followed the reviewer's advice and determined whether *Isl1*⁺/*Nkx2-5*⁺ CPCs show a common gene expression signature that differs from cells that are either *Isl1*⁺ or *Nkx2-5*⁺ along with pseudotime analysis of *Isl1*⁺/*Nkx2-5*⁺ CPCs (Fig. 3d, e; Fig. 4). Analysis of scRNA-seq data revealed that there are only few *Isl1*⁺/*Nkx2-5*⁺ cells, which is consistent with the immunofluorescence staining of *Isl1*^{+/nGFP}/*Nkx2-5*-emGFP double positive mice. In general, we detected fewer cells with *Isl1* and *Nkx2-5* transcripts in the *Isl1* lineage (defined by *Isl1*^{+/nGFP}) compared to the *Nkx2-5* lineage. In addition, the expression level of *Nkx2-5* was relatively low. In contrast, the *Nkx2-5* lineage (defined by *Nkx2-5*-emGFP) contained more double positive cells and the expression level of *Isl1* was higher. When we took the *Isl1* diffusion map and positioned *Nkx2-5* expressing cells onto the map, we were able to classify several categories: (1) *Isl1*⁺/*Nkx2-5*⁻, (2) *Isl1*⁻/*Nkx2-5*⁺, and (3) cells with varying levels of both *Isl1* and *Nkx2-5*. Co-expressing cells were color-coded, more green when *Isl1* was expressed at higher levels and more red when *Nkx2-5* was expressed at higher levels, (4) double negative cells. In Fig. 3d, we only considered cells from the *Isl1*-lineage and crossed out cells from the *Nkx2-5* lineage. In Fig. 3e, cells from *Nkx2-5* lineage were colored and cells from *Isl1*-lineage were crossed out. Consistently, *Nkx2-5* is only expressed in cells that fit on the trajectory towards cardiomyocytes, irrespectively of whether the cells co-express *Isl1*. We can even detect a switch-like expression at the end of the trajectory. In contrast, cells are *Isl1* positive but *Nkx2-5* negative (all cells are labeled green) during the transition state. Early cells in the *Nkx2-5* trajectory also express *Isl1*, but to a lower extent, suggesting that early *Nkx2-5* expression primes CPCs towards a cardiac fate. Furthermore, *Nkx2-5* primed cells seem to take a shortcut to cardiomyocytes, since they do not overlap with the *Isl1* transition state.

The detection of double negative cells (i.e. cells, in which neither *Isl1* nor *Nkx2-5* transcripts were detected), mostly at early (E7.5) stages, seems counterintuitive, since all CPCs were isolated based on reporter gene activity driven by the *Isl1* or *Nkx2-5* promoters. However, such a phenomenon is not unusual and most likely reflects transcriptional fluctuations that are not accurately represented by the GFP proteins, which are more stable than the mRNAs. Previous studies demonstrated that mRNAs of genes controlling cell-type specific functions in a limited time window usually have short life spans (<2-3h) and show fluctuations of transcription [2]. Double negative cells, probably reflecting such fluctuations were primarily detected in early CPCs progenitors when transcription of *Isl1* and *Nkx2-5* becomes established. Of note, removal of double negative cells did not affect the outcome of the clustering. We therefore did not discard such cells, since we believe they reflect the real transcriptional dynamics/fluctuation of *Isl1* and *Nkx2-5* and contribute to the heterogeneity.

Direct comparison of *Isl1*⁺, *Nkx2-5*⁺CPCs or *Isl1*⁺/*Nkx2-5*⁺ cells did not yield a distinct cluster that would not fit into *Isl1*⁺ or *Nkx2-5*⁺ clusters further suggesting that double positive cells do not follow a distinct fate but represent a temporary state along the trajectories (Fig. 3d). To further highlight the differences between *Isl1* and *Nkx2-5* expressing cells, we performed in situ hybridization of a few candidate genes, which were found to be differentially expressed in our bioinformatics analysis (Fig. 4).

3. How is ATACseq limited by the fact that it was applied to bulk populations which contain mixtures of CPC and more differentiated cells that change over time. How might this affect the results and conclusions?

Response: The reviewer is obviously right that the use of bulk preparations of cells for ATAC-seq limits the usefulness of the data. Therefore, we have performed chromatin accessibility assays in single *Isl1*⁺ CPCs (single cell ATAC-seq or scATAC-seq). The data set provides additional important insights into the heterogeneity of CPCs. Importantly, the single cell chromatin accessibility profiles resemble the transcriptional heterogeneity of *Isl1*⁺ CPCs. We think that we have greatly strengthened our conclusions by confirming the existence of different CPC sub-population via scATAC-seq. The scATAC-seq essentially recapitulated the cardiomyocyte and endothelial lineage bias demonstrated by our scRNA-seq and enabled us to explore the TF binding dynamics in different developmental trajectories (Figs. 6, 7).

4. It is difficult to keep track of the different populations identified by tSNE analysis as the narrative unfolds. It would be help full to further label figures to indicate the key populations under discussion and their qualifiers – eg. EC, CM, CPCearly/late etc.

Response: We thank the reviewer for pointing out this shortcoming. We have added names to clusters using qualifier names such as EC (endothelial cells), CM (cardiomyocyte), CPC early/intermediate/late together with numbers, which was also suggested by Reviewer #1 (Point 9) and Reviewer #2 (Point 3). We hope that this type of labeling help readers to keep track of the narrative of the manuscript.

5. Page 5. I am confused by the assignment of Nkx2-5 cluster 3 as a CPC being from E7.5. This does not seem to fit with the pseudotrajectory arrow in Fig. 2A and the finding that this cluster express higher levels of CM markers than the other later clusters in particular cluster 1 (Fig. 1f – cTnC, Sma).

Response: We thank the reviewer for pointing out this mistake, which was also noticed by Reviewer #2 (Point 1). The confusion is due to a typo resulting in wrong labeling of the two clusters, which we have corrected.

6. It should be acknowledged that Nkx2-5 is also expressed in a subset of endocardial cells. These are not captured, likely because Nkx2-5 and GFP are not robustly expressed there.

Response: We agree with the reviewer that we might have missed subsets of endocardial cells that express Nkx2-5 only transiently or at low levels. We have added a corresponding statement.

7. Page 6 para 3. It says here that Nkx2-5 clusters 3 and 1 correspond to E8.5 and E9.5 respectively. See above. Is this a mistake?

Response: We thank the reviewer for addressing this issue, which was also recognized by reviewer #2. Our analysis clearly demonstrates that development of Isl1⁺ or Nkx2-5⁺ cells is not synchronized at any given time point. With other words: there are already some cells at E8.5 that are more mature than others. The same is true for E9.5 cells: some cells are less mature than others, resembling more E8.5 cells. This phenomenon can give rise to the seemingly paradox situation pointed out by the reviewer. In our view, the temporal stages at which the cells were isolated do not necessarily reflect the “true” developmental stages but allow inclusion of cells that are either a bit advanced or behind the developmental schedule.

8. Does it seem surprising that clusters seem very discrete and that cells bridging clusters are not generally evident. Can discrete clusters be claimed given so few cells profiled? Are there insights from the replication experiment with iCell8 platform.

Response: Figure 1c shows the results of clustering after the computational analysis. The clustering algorithm allows outliers (i.e. cells that do not belong to a cluster). Such cells, which are typically located between clusters, are usually filtered out and are not shown. We have revised the figures, which also now include outliers that are localized at the edges/borders of clusters (as shown by grey cross in the figures). The Wafergen data identified more cells outside of the clusters (Supplementary Fig. 5) but we did not include a detailed description, since the complexity of the paper is already rather high.

9. Page 7: what is meant by the conclusion that CPS are not synchronized at different embryonic stages but follow individual developmental traits?

Response: This issue relates to point #7 above and simply means that not ALL Isl1⁺ or Nkx2-5⁺ cells at a given temporal stage have reached the same degree maturity. The single cell data clearly indicate the time axis is not precisely reflecting the true developmental stage. Of course, we do not want to claim that each cell entirely follows an individual trait in a strict sense. Different cells seem to respond differentially to signaling cues and the activity of specific signaling cues might vary within the embryo resulting in heterogeneity of cells. We have rephrased the sentence.

10. Why would I_c(c) values be higher or equal in transition states than in differentiated states. Further rationale and assumptions need to be incorporated into this section.

Response: It is generally assumed that differentiated cells are in a stable state driven by a dominant regulatory network, which restrains transcriptional noise and keeps gene expression symmetrically centered around the population means. Cell fate transitions correspond to a switch between distinct stable states, initiated by a gradual destabilization of the initial stable state (e.g. triggered by exogenous signals such as differentiation cues). Gene expression of transitioning cells is not as constrained as in stable cells, meaning that transcriptional noise increases upon transition resulting in increased heterogeneity. Similarly, cell-to-cell correlations are high in stable cell states, mostly driven by characteristic expression patterns that define the specific cell type. Gene-to-gene correlation however is low, because gene expression only fluctuates around their population mean and gene expression does not change in a coordinated manner. During transition between different states, cell-to-cell correlation decreases as cells leave stable states. Interestingly, gene-to-gene correlation increases, since transition-specific expression programs initiate a coordinated change of gene expression in each cell. The $Ic(c)$ value is the ratio between absolute gene-gene (numerator) and cell-cell correlation (denominator). It is therefore higher in transitioning cells, when high gene-gene correlation (coordinated expression changes, numerator) determines the ratio.

We cannot elaborate extensively on these issues in the current paper. Instead we would like to refer the reader to published work by Bargaje et al. [4] and Mojtahedi et al. [5], which was cited in our manuscript. In addition, we would like to emphasize that a similar approach was chosen by Mohammed et al. [6], who used transcription noise to determine cell transition states. We have added an additional sentence to highlight the references describing the underlying theories.

11. Page 13 sentence 2. Add “compare to WT”? But how can this be determined given that the inactivation embryos were analysed only at one time point – E9.5?

Response: The reviewer is right that the E9.5 *Isl1* mutant CPCs were compared to E9.5 WT cells. ATAC-seq was initially performed on FACS-sorted *Nkx2-5*⁺ and *Isl1*⁺ WT cardiac progenitor cells and on E9.5 *Isl1*⁺ mutant CPCs. We think that it is completely legitimate to compare mutant cells from one developmental stage to WT cells from the same and other developmental stages. Nevertheless, we performed several additional experiments to alleviate the reviewer's concerns. (i) we performed scATAC-seq of *Isl1*⁺ CPCs at E8.5 and E9.5. (ii) we compared ATAC-seq data from E9.5 *Isl1*⁺ mutant CPCs to data from E9.5 and E8.5 *Isl1*⁺ WT CPCs, which had been separated into two distinct subpopulations using a CD31 antibody (*Isl1*⁺CD31⁻ and *Isl1*⁺CD31⁺ cells). The *Isl1*⁺CD31⁻ cells represent the trajectory towards cardiomyocytes whereas *Isl1*⁺CD31⁺ cells represent the endothelial trajectory as indicated by the scRNA-seq data. Strikingly, inactivation of the *Isl1* gene did not cause substantial changes compared to *Isl1*⁺CD31⁻ CPCs at E8.5 or E9.5 (Fig. 9a & b). Loss of *Isl1* led to more robust closing than opening of chromatin regions compared to *Isl1*⁺CD31⁺ CPCs at E8.5 and E9.5 suggesting that *Isl1* is required to leave the attractor state characterized by an open chromatin organization (Fig. 9a, b). Taken together, these results indicate that *Isl1* mutant CPCs exhibit an epigenomic profile that resembles developing cardiomyocytes and differs from endothelial cells.

12. Page 13: “...92% of the peaks...presumably representing enhancer regions affecting progenitor cell decisions.” Unclear.

We assume that the reviewer refers to the sentence “Annotation of opening and closing peaks by GREAT analysis showed that 96.2% of the peaks (1,256) located in distal regions (> ±5 kb to the TSS sites) presumably representing enhancer regions affecting cardiac progenitor cell decisions (Fig. 6c).” We have now defined more precisely the criteria used to discriminate proximal and distal regulatory regions in both the main text and the Methods sections (see below).

“The proximal and distal peaks were defined by the distance of differential ATAC-seq peaks towards annotated promoters (Gencode annotation): peaks at least 2.5 kb away from transcriptional start sites were defined as distal peaks, peaks closer to transcriptional start sites were defined as proximal peaks.”

In addition, we have thoroughly revised the whole paragraph due to inclusion of additional ATAC-seq and scATAC-seq data.

13. Page 13: “We concluded that *Isl1* acts together with *Mef2* factors but prevents binding of Forkhead factors...” Not sure this fits with what was said previously.

Response: Since we performed additional ATAC-seq and scATAC-seq experiments and increased the quality of the ATAC-seq data, we were able to conduct a more refined motif analysis. We have thoroughly revised the incriminated paragraph and eradicated all inconsistencies.

14. Page 14: “..forced expression of Nkx2-5 resulted in a dramatic increase of accessible chromatin sites at E9.5 compared to E12.5..” Don’t the authors mean E12.5 relative to E9.5? See above.

Response: We are sorry for the confusion, which probably resulted from the global comparison of Nkx2-5 overexpressing cells to Nkx2-5 overexpressing cells from a different stage (E9.5 Nkx2-5 OE versus E12.5 Nkx2-5 OE) but also of Nkx2-5 overexpressing cells to WT cells (E9.5 Nkx2-5 OE versus E9.5 Nkx2-5). We indeed saw a relative increase of chromatin opening in Nkx2-5 OE cells at E9.5 compared to Nkx2-5 OE cells at E12.5 (Suppl. Fig. 14e). However, the heatmap shown in (Suppl. Fig. 14f) paints a more detailed picture: Clusters 1 and 4 show a clear increase of chromatin opening at E9.5 after Nkx2-5 overexpression, which disappeared at E12.5, while clusters 2 and 3 showed closure of chromatin at E9.5 after Nkx2-5 overexpression but reversal of that phenomenon at E12.5. From these data we concluded that the increased opening due to Nkx2-5 happened only transiently but was not sustained probably due to CPC differentiation. We have improved description of the data in the main text to cope with this issue.

Reviewer #4 (Remarks to the Author):

This is a nicely written and comprehensive study about cardiac progenitor cell fate decisions. Overall, I have a positive view about this manuscript, but it needs some improvements to be acceptable for publishing. In particular, there are a number of conclusions that are not supported by the experiments/analysis performed. Here are my specific comments:

1) It remains unclear how many embryos per stage were used in experiments shown in Figure 1 and 2. If the trajectory and all the subsequent analyses were generated based on the data from one embryo from each stage, then authors should include data from more embryos using the same experimental approach so that data can be combined and reanalysed. This would give confidence in the robustness of the conclusions.

Response: The CPCs were sampled at different timepoints using 403 embryos in total. EACH embryo was accurately staged based on the number of somites to allow precise matching of different developmental stages. We have updated the information in the Methods section:

“In total, 403 embryonic hearts were harvested for scRNA-seq, scATAC-seq, bulk RNA-seq and bulk ATAC-seq experiments. Each embryo was accurately staged based on the number of somites to allow precise matching of different developmental stages [...]”

We can provide information about the numbers of pooled embryonic hearts that were used for each experiment, but we do not think that this make much sense, since the yield of CPCs did not correlate strictly to the number of hearts that were used for the different FACS runs. Of course, it is not possible to assign individual cells to specific embryos after pooling. We think that the number of cells, which were analyzed in each experiment, is more informative.

Since the data generated from three different developmental stages were combined how did the authors deal with batch effects? Especially in the case of Nkx-2 clustering where it appears that two out of three clusters contain cells from distinct developmental stages.

Response: We examined batch effects by plotting Fluidigm C1 runs to the clusters (see the plot as below). For each run, at least two of Fluidigm IFCs were used to collect cells, and after removal of doublets and reverse transcription, cDNAs were synthesized, multiplexed with ~96 barcodes and sequenced on a Nextseq 500 in one run. We found that cells from the same (temporal) developmental stage located into different clusters, which clearly argues that different clusters are not generated by batch effects. Of course, some runs primarily located into single clusters but this correlated with distinct developmental stages. For example, cluster 3 of Nkx2-5 comes mainly from the C1run3, which corresponds to the fact that cells were sampled at E9.5 cells and this timepoint was only run once. Another, more important argument comes from the use of the Wafergen platform. Here, we analyzed cells from three different developmental timepoints in one experiment, which was possible since the Wafergen chip contains 5,148 wells allowing loading of Nkx2-5⁺ cells from different stages, thereby enabling us to exclude any batch effects. The Wafergen experiment uncovered the same 3 subpopulations as the Fluidigm C1 experiment (Supplementary Fig. 5). Therefore, we concluded that batch effects have only a minimal impact on our results.

2) Authors stated the following:

“Projection of *Isl1*-KO single cells on the trajectory of the developing SHF revealed that *Isl1*-KO cells are stalled/trapped in the previously identified stable attractor state (Fig. 3b).”

This is a 2D projection. The *Isl1*-KO cells might be perpendicular (or far away) to the wt cells in the 3D space. Authors should show data in 3D so that it is clear that they are really occupying the same attractor state.

Response: We have replaced the 2D plot in the previous Fig. 3b with a 3D plot (now showing in Fig. 5b) as requested. It is clearly visible in the 3D plot that *Isl1*-KO cells localize close to the attractor state, mostly in clusters 3 and 4. Importantly, *Isl1*-KO cells are far away from the endothelial and cardiomyocyte branches, which suggests that the inactivation of *Isl1* prevents differentiation of CPCs and locks them in the attractor state (Fig. 5b). Furthermore, we combined data from E7.5, E8.5 and E9.5 *Isl1*⁺ with E9.5 *Isl1* mutant cells and repeated the clustering (Fig. 5b). As expected, E9.5 *Isl1* mutant cells intermixed mostly with clusters 3 and 4.

3) Authors stated the following:

“Our pseudotime-based analysis of developmental trajectories revealed one continuous trajectory of *Nkx2-5*⁺ CPC suggesting that *Nkx2-5*⁺ cells are exclusively committed to become cardiomyocytes. In contrast, the trajectory of *Isl1*⁺ CPC bifurcated into two distinct orientations (endothelial cells and cardiomyocytes), suggesting the existence of a transition state, which separates multipotency of *Isl1*⁺ CPC from acquisition of distinct cellular identities (Fig. 2b).”

It is not correct to make such a statement just based on the way trajectory looks like. These two trajectories (*Nkx2-5*⁺ and *Isl1*⁺) look very similar (in terms of overall shape) and only the subsequent analysis revealed the differences between them. Therefore, this part should be rewritten and conclusions should be made later on in the manuscript when all the evidence for such a claim has been laid out.

Response: The reviewer is obviously right. The CPCs were sampled at three different timepoints, which is why we anticipated the pseudotime trajectories. We now make this claim after delineation of the transition states using the lc(c) value calculation.

An important point here is that the lack of cells with the endothelial identity in Nkx2-5+ trajectory is not the proof of their unilineage potential i.e. committed state. As authors noted themselves lack of expression of Nkx2-5 in the endothelial cells might have prevented their perspective sorting. In fact, there is a compelling evidence that some of Nkx2-5+ cells are multipotent. First, the lineage tracing studies have shown the multipotency of Nkx2-5+ cells and second the endogenous expression of Nkx2-5 in cluster 3 is close to zero.

Thus, the authors should not make such claims unless they are willing to do functional experiments to prove that indeed Nkx2-5+ are already committed. Instead, I would suggest that they point out limitations of their system (e.g. down regulation of Nkx2 in endothelial cells) and move on to show the functional relevance of this gene in the cardiomyocyte differentiation. This has been nicely demonstrated in their Nkx2-5 expression experiment.

We thank the reviewer for the comment. Nkx2-5+ cells clearly also contribute to other lineages. We certainly do not want to claim that Nkx2-5+ CPCs are exclusively unipotent and have therefore revised the wording. Instead, we emphasize Nkx2-5 is required and sufficient to resolve the multipotent differentiation capacity of CPCs and drives cells along the cardiomyocyte developmental trajectory.

5) Authors stated the following:

Since our scRNA-seq analysis at E9.5 indicated arrest of Isl1 mutant CPC development, essentially converting the transcriptional profile of E9.5 into an E8.5 state (Fig. 3), we compared the landscape of chromatin accessibility of both populations. Surprisingly, we found that E8.5 Isl1+ cells and E9.5 Isl1 mutant cells show significantly different open chromatin signatures (432 opening and 728 closing), although they share similar positions at the developmental pseudotime trajectory (Fig. 6a), which indicates that Isl1-dependent changes in chromatin accessibility occur ahead of transcriptional divergence.

I do not see the evidence that isl1 mutant CPC have been transcriptionally converted from E9.5 to E8.5. First, although isl1KO cells cluster together their relationship to isl1+ cells cannot be assessed based on the 2D projection that authors show. It would be more appropriate to combine data from E7.5, E8.5 and E9.5 Isl1+ with E9.5 Isl1 mutant and perform clustering (similar to the one in Figure 1). The expectation is that E8.5 Isl1+ cells and E9.5 Isl1 mutant cells would cluster together (if they are transcriptionally similar they should be intermixed). The actual transcriptional differences between E8.5 Isl1+ cells and E9.5 Isl1 mutant cells could be assessed by performing differential expression analysis. Second, isl1+ cells from E9.5 and E8.5 are intermixed in pseudotime and sit close to each other at the beginning of the trajectory. On the page 5 authors stated: "Stage-dependent clustering was less evident for the five Isl1+ subpopulations....". Therefore, I disagree with the logic to compare open chromatin signatures of E8.5 Isl1+ cells and E9.5 Isl1 mutant cells and the statement that they share similar positions at the developmental pseudotime trajectory.

Response: As already explained above in the response to point #2, we have replaced the 2D plot in the previous Fig. 3b with a 3D plot (now showing in Fig. 5b) as requested. It is clearly visible in the 3D plot that Isl1-KO cells localize close to the attractor state, mostly in clusters 3 and 4. Importantly, Isl1-KO cells are far away from the endothelial and cardiomyocyte branches, which suggests the inactivation of Isl1 prevents differentiation of CPCs and locks them in the attractor state (Fig. 5b). Furthermore, we combined data from E7.5, E8.5 and E9.5 Isl1+ with E9.5 Isl1 mutant cells and repeated the clustering (Fig. 5b). As expected, E9.5 Isl1 mutant cells intermixed mostly with clusters 3 and 4.

In addition, we performed additional ATAC-seq experiments: (i) we performed scATAC-seq of Isl1+ CPCs at E8.5 and E9.5. (ii) we compared ATAC-seq data from E9.5 Isl1+ mutant CPCs to data from E9.5 and E8.5 Isl1+ WT CPCs, which had been separated into two distinct subpopulations using a CD31 antibody (Isl1+CD31- and Isl1+CD31+ cells). The Isl1+CD31- cells represent the trajectory towards cardiomyocytes whereas Isl1+CD31+ cells represent the endothelial trajectory as indicated by the scRNA-seq data. Strikingly, inactivation of the *Isl1* gene did not cause substantial changes compared to Isl1+/CD31- CPCs at E8.5 or E9.5 (Fig. 9a & b). Loss of Isl1 led to more robust closing than opening of chromatin regions compared to Isl1+/CD31+ CPCs at E8.5 and E9.5 suggesting that Isl1 is required to leave the attractor state characterized by an open chromatin organization (Fig. 9a, b). Taken together, these results indicate that Isl1 mutant CPCs exhibit an epigenomic profile that resembles developing cardiomyocytes and differs from endothelial cells.

Minor point – authors should change the colours of cells at different stages of development. At the moment it is hard to see distribution of cells in pseudotime because different shading of the same colour is used.

Response: We have changed the colors in the plots to improve visualization of the distribution of cells as requested.

Reference

1. Yuan, X., et al., *Disruption of spatiotemporal hypoxic signaling causes congenital heart disease in mice*. J Clin Invest, 2017. **127**(6): p. 2235-2248.
2. Shah, S., et al., *Dynamics and Spatial Genomics of the Nascent Transcriptome by Intron seqFISH*. Cell, 2018. **174**(2): p. 363-376 e16.
3. Shen, Y., et al., *A map of the cis-regulatory sequences in the mouse genome*. Nature, 2012. **488**(7409): p. 116-20.
4. Bargaje, R., et al., *Cell population structure prior to bifurcation predicts efficiency of directed differentiation in human induced pluripotent cells*. Proc Natl Acad Sci U S A, 2017. **114**(9): p. 2271-2276.
5. Mojtahedi, M., et al., *Cell Fate Decision as High-Dimensional Critical State Transition*. PLoS Biol, 2016. **14**(12): p. e2000640.
6. Mohammed, H., et al., *Single-Cell Landscape of Transcriptional Heterogeneity and Cell Fate Decisions during Mouse Early Gastrulation*. Cell Rep, 2017. **20**(5): p. 1215-28.

REVIEWERS' COMMENTS:

Reviewer #1 (Remarks to the Author):

I thank the authors for performing all requested experiments and the detailed response to my questions and concerns. I highly recommend the revised version for publication. It provides great insights into early cardiac lineages on the single cell level.

Reviewer #3 (Remarks to the Author):

The revised manuscript by Jia and Braun et al. is much improved. The incorporation of single cell ATACseq data in particular is a strong advance. The point by point responses to reviewers' comments have been written carefully and in my view address the technical and philosophical issues raised. I am very happy to support publication. The paper is a very significant resource and reveals new insights into progenitor - differentiated cell transitions supported by deep and unbiased data.

Reviewer #4 (Remarks to the Author):

I think that the authors adequately addressed all concerns that I had.